# Partial Matrix Completion

**Elad Hazan** *
Princeton University
Google DeepMind

**Adam Tauman Kalai** *
Microsoft Research

**Varun Kanade** *
University of Oxford

**Clara Mohri** *
Harvard University

**Y. Jennifer Sun** *
Princeton University
Google DeepMind

## Abstract

The matrix completion problem aims to reconstruct a low-rank matrix based on a revealed set of possibly noisy entries. Prior works consider completing the entire matrix with generalization error guarantees. However, the completion accuracy can be drastically different over different entries. This work establishes a new framework of *partial matrix completion*, where the goal is to identify a large subset of the entries that can be completed with high confidence. We propose an efficient algorithm with the following provable guarantees. Given access to samples from an unknown and arbitrary distribution, it guarantees: (a) high accuracy over completed entries, and (b) high coverage of the underlying distribution. We also consider an online learning variant of this problem, where we propose a low-regret algorithm based on iterative gradient updates. Preliminary empirical evaluations are included.

## 1 Introduction

In the classical matrix completion problem, a subset of entries of a matrix are revealed and the goal is to reconstruct the full matrix. This is in general impossible, but if the matrix is assumed to be low rank and the distribution over revealed entries is uniformly random, then it can be shown that reconstruction is possible. A common application of matrix completion is in recommendation systems. For example, the rows of the matrix can correspond to users, the columns to movies, and an entry of the matrix is the preference score for the user over the corresponding movie. The completed entries can then be used to predict user preferences over unseen movies. The low rank assumption in this case is justified if the preference of users over movies is mostly determined by a small number of latent factors such as the genre, director, artistic style, and so forth.

However, in many cases, it is both infeasible and unnecessary to complete the entire matrix. First consider the trivial example of movie recommendations with users and movies coming from two countries $A$ and $B$, where each user rates random movies only from their country. Without any cross ratings of users from country $A$ on movies from country $B$ or vice versa, it is impossible to accurately complete these entries. A solution here is straightforward: partition the users and movies into their respective groups and then complete only the part of the matrix corresponding to user ratings of movies from their own country.

In reality, many users have categories of movies with few or no ratings based on genre, language, or time period. For such users, it is difficult to accurately complete unrated movie categories which they do not like to watch from those which they have not even been exposed to. Thus it may be preferable to abstain from making rating predictions for such users on these unpredictable categories. This is

---

* Authors ordered alphabetically

37th Conference on Neural Information Processing Systems (NeurIPS 2023).

further complicated by the fact that such categories are not crisply defined and relevant categories may vary across user demographics like country and age.

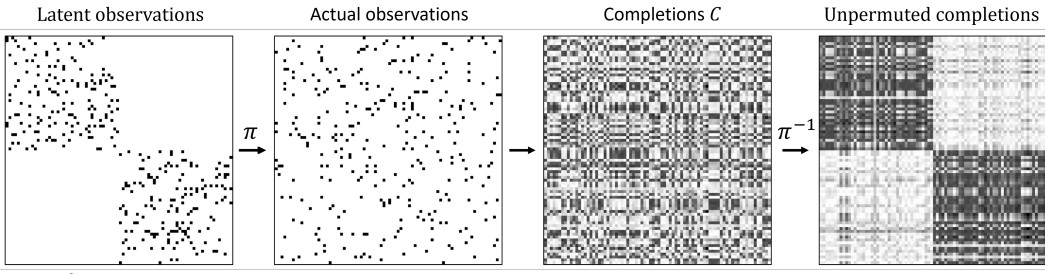

Figure 1: A simple example where some users rate only comedies and some rate only horror movies. However, these two groups are unknown and the rows and columns are permuted according to $\pi = \pi_C \circ \pi_R$. The partial completions made by our algorithm, shown on simulated data, correspond to completing the ratings for the half of the matrix as one would like. Note that our algorithms can handle *arbitrary* revealed subsets, not only stochastic block models. The experimental setup and more preliminary experiments are described in Section 6.

This motivates a more general question: can we identify a subset of the matrix that can be completed by the existing observations with high confidence? We call this problem **partial matrix completion**. Let $M^\star$ be a target matrix. A *non-convex* formulation would be to find a matrix $\hat{M}$ and a set of entries $C \subseteq [m] \times [n]$ which maximizes:

$$\max_{C \subseteq [m] \times [n]} |C| \text{ s.t. } \frac{1}{|C|} \sum_{(i,j) \in C} (M_{ij}^\star - \hat{M}_{ij})^2 \leq \varepsilon. \tag{1}$$

This is non-convex because the set $C$ is discrete and also because the choice of $\hat{M}$ interacts $C$. Fortunately, there are already several matrix completion algorithms which output $\hat{M}$ that guarantee low "generalization error" with respect to future examples *from the same distribution of entries* as those from the training distribution, though they are not accurate over the entire matrix. We will fix any one of these completions $\hat{M}$.

We thus relax the goal of partial matrix completion to identify a matrix $C$ which gives a confidence score for each entry between 0 and 1. A value of 0 indicates that this entry should not be completed, whereas 1 indicates we have absolute confidence. We call this matrix the **confidence matrix**. Ideally, we would like that the confidence matrix to have a value of 1 on all entries that are well supported by the observation distribution.

We formalize this problem through the following definitions. The precise definitions of some of the terms will be introduced in subsequent sections. Let $M^\star$ be a target matrix and $\hat{M}$ be a completion given by a matrix completion algorithm that guarantees low generalization error over future examples from the same entry distribution (formal definition given in Def. 8), and let $\varepsilon > 0$. The *coverage* of $C$ is defined to be $\|C\|_1 \stackrel{\text{def}}{=} \sum_{ij} |C_{ij}|$.

**Definition 1** (Partial matrix completion problem). Let $M^\star$ be a target matrix and $\hat{M}$ be a full completion (Def. 8), and $\varepsilon > 0$ be an error tolerance. Find a confidence matrix $C \in [0,1]^{m \times n}$ with maximal coverage, i.e.

$$\underset{C \in [0,1]^{m \times n}}{\arg \max} \|C\|_1 \text{ s.t. } \frac{1}{\|C\|_1} \sum_{i \in [m], j \in [n]} C_{ij} (M_{ij}^\star - \hat{M}_{ij})^2 \leq \varepsilon.$$

The matrix $C$ that solves the above optimization problem has the following property: $C_{ij}$ indicates whether the $(i, j)$-th entry shall be completed, and a fractional value can be interpreted as either a fractional completion or a probability of completion. $C$ provides the largest coverage among all that satisfy the property that the average error is at most $\varepsilon$ over the completed entries.

**Challenges of partial matrix completion.** The above formulation is intuitive and convex, but notice that we do not know $M^\star$, and thus cannot apply convex optimization directly.

Key challenges are illustrated by two intuitive, but naïve approaches. First, consider completing just those matrix entries $(i, j)$ where all rank-$k$ completions have nearly the same value $\hat{M}_{ij}$. Even in a simple case where all revealed entries are 1, any single entry could be $\pm 1$ for some rank 2 completion of the matrix. Based on this, one often will not be able to complete any entries whatsoever.

Second, consider the generalization of our previous example where the sampling distribution is uniform over an *arbitrary* subset $U$ of $m \times n$ possible indices. For simplicity, think of $|U| = mn/2$ as covering half the matrix. It follows from previous generalization bounds on matrix completion that one could accurately complete all entries in $U$. However, in general $U$ is unknown and arbitrarily complex. The second naïve approach would be to try to learn $U$, but this requires $\approx |U| = mn/2 \gg m + n$ observations.

## 1.1 Our contribution

We present an inefficient algorithm and its efficient relaxation to solve the partial matrix completion problem with provable guarantees over arbitrary sampling distributions. In particular, if $\mu$ is uniform over some arbitrary subset $U \subseteq [m] \times [n]$ of a rank-$k$ matrix $M^\star \in [-1, 1]^{m \times n}$, the theoretical results obtained in this work have the following implications. The first, is an inefficient algorithm that solves the partial matrix completion problem for uniform distribution:

**Corollary 2** (Inefficient algorithm, uniform sampling distribution). *For any $\varepsilon, \delta > 0$, with probability $\geq 1 - \delta$ over $N \geq c_0 \cdot \frac{k(m+n)+\log(1/\delta)}{\varepsilon^2}$ revealed entries from $M^\star$ drawn from $\mu$, for some constant $c_0$, Algorithm 1 (Sec. 3) outputs $\hat{M}$ and $C \in [0, 1]^{m \times n}$, such that*

$$(1): \ \|C\|_1 \geq |U|,$$
$$(2): \ \frac{1}{\|C\|_1} \sum_{i \in [m], j \in [n]} C_{ij}(M_{ij}^\star - \hat{M}_{ij})^2 \leq \varepsilon.$$

Next, we describe the guarantees of an efficient algorithm 2. It has a worse sample complexity, up to polynomial factors in $k, \varepsilon$. The advantage is that it runs in polynomial time. Otherwise, its guarantees are similar to that of the first algorithm.

**Corollary 3** (Efficient algorithm, uniform sampling distribution). *For any $\varepsilon, \delta > 0$, with probability $\geq 1 - \delta$ over $N \geq c_0 \cdot \frac{k^2}{\varepsilon^4} \cdot (k(m + n) + \log(1/\delta))$ revealed entries from $M^\star$ drawn from $\mu$, for some constant $c_0$, Algorithm 2 (Sec. 4) outputs $\hat{M}$ and $C \in [0, 1]^{n \times m}$, such that*

$$(1): \ \|C\|_1 \geq |U|,$$
$$(2): \ \frac{1}{\|C\|_1} \sum_{i \in [m], j \in [n]} C_{ij}(M_{ij}^\star - \hat{M}_{ij})^2 \leq \varepsilon.$$

Corollary 2 is a simplification of Theorem 10, and Corollary 3 is a simplification of Theorem 11.

The results can be extended to the setting where the observed entries are noisy with zero-mean bounded noise. A remarkable feature of our algorithms is that once the full completion $\hat{M}$ is obtained using existing procedures, they only rely on the *locations* of observed entries and not the values. Thus, the decisions regarding which entries to complete, i.e., to add to $C$ is completely agnostic to the actual values of the revealed entries. Our framework is able to handle *arbitrary* sampling distributions, which captures scenarios such as overlapping groups or idiosyncratic rating habits.

**The online setting.** We formulate the partial matrix completion problem in the online learning setting and propose an iterative gradient based online algorithm with provable guarantees. Corollary 4 is a special case of Theorem 22. A simple simulation of the online algorithm in this special case of uniform sampling over a subset of entries is demonstrated in Fig. 1.

**Corollary 4** (Online algorithm, uniform sampling distribution). *Suppose $M^\star \in [-1, 1]^{n \times n}$ is a bounded matrix with max-norm (Eq. 2) bounded by $K$. The sampling distribution $\mu$ is uniform over a fraction $0 < c \leq 1$ of the $n^2$ entries. For any $\delta > 0$, after $T = \tilde{O}(\delta^{-2}K^2 n)$ iterations, the*

*confidence matrix $C \in [0,1]^{m \times n}$ output by the online algorithm ODD satisfies with probability at least $1 - c_1 \exp(-c_2 \delta^2 T)$ for some universal constants $c_1, c_2 > 0$,*

$$(1): \quad \|C\|_1 \geq (c - \delta^{1/6})n^2,$$

$$(2): \quad \frac{1}{\|C\|_1} \sup_{M \in \mathcal{V}} \sum_{i \in [m], j \in [n]} C_{ij}(M_{ij} - M_{ij}^\star)^2 \leq \frac{\delta^{1/6}}{c - \delta^{1/6}},$$

*where $\mathcal{V}$ is the set of all matrices $M \in [-1,1]^{n \times n}$ with max-norm bounded by $K$ satisfying that $\mathbb{E}_{i,j \sim \mu}[(M_{ij} - M_{ij}^\star)^2] \leq \delta$.*

## 1.2 Related work

**Matrix completion and recommendation systems.** The common approach to collaborative filtering is that of matrix completion with trace norm minimization [Srebro, 2004]. It was shown that the trace norm regularization requires $\Omega((n + m)^{3/2})$ observations to complete the matrix under arbitrary distributions, even when the rank of the underlying matrix is constant, and this suffices even in the more challenging agnostic online learning setting [Hazan et al., 2012a, Shamir and Shalev-Shwartz, 2014]. Srebro and Shraibman [2005] study matrix completion with the max-norm, which behaves better under arbitrary distributions, namely for low-rank matrices, $\tilde{O}(n + m)$ observations suffice.

**Matrix completion and incoherence assumptions.** A line of works in the matrix completion literature considered the goal of finding a completion of the ground truth matrix with low Frobenius norm error guarantee [Candès and Recht, 2009, Candès and Tao, 2010, Keshavan et al., 2010, Gross, 2011, Recht, 2011, Negahban and Wainwright, 2012, Jain et al., 2013, Chen et al., 2020, Abbe et al., 2020]. Such guarantee is strong but usually requires two restrictive assumptions on sampling distribution of observations and the ground truth matrix structure: (1) the sampling of observations is uniform across all entries, and (2) the ground truth matrix $M^\star$ satisfies some incoherence conditions. The incoherence condition is an assumption imposed on the singular vectors of $M^\star = U \Sigma V^\top$, which ensures the vectors $u_i, v_j$'s to be sufficiently spread out on the unit sphere. The main reason the incoherence condition is necessary in establishing meaningful guarantees in low-rank matrix completion is that without such assumptions, there might exist multiple low-rank matrices that could explain the observed entries equally well but differ substantially on unobserved entries. When the incoherence condition is satisfied, it implies that the observed entries capture enough information about the low-rank structure, making it possible to recover the original matrix with sufficiently well. However, uniform sampling and incoherence assumptions *do not* hold in many realistic scenarios.

Subsequently, to circumvent these restrictive assumptions, another line of work evaluates the *generalization error* as an alternative metric for completion performance [Srebro et al., 2004, Shamir and Shalev-Shwartz, 2014]. To formalize, consider the task of completing $M^\star$ with an arbitrary observation sampling distribution $\mu$ over $[m] \times [n]$. This can be conceptualized as predicting a hypothesis mapping from the domain $[m] \times [n]$ to the codomain $[-1, 1]$. The goal is to characterize the guarantees on the expected prediction error, quantified for a given completion $\hat{M}$ as $\mathbb{E}_{(i,j) \sim \mu}[(\hat{M}_{ij} - M_{ij}^\star)^2]$. When $\mu$ is the uniform distribution over $[m] \times [n]$, the generalization error bound translates to a guarantee over the average error across all entries. However, if $\mu$ is arbitrary (e.g. supported over a fraction of the entries), then no guarantee could be established for entries that lie in the complement of the support of $\mu$. This prohibits the use of $\hat{M}$ in settings where abstention is imperative.

Our work takes an alternative approach to this fundamental limitation in matrix completion. The focus of our work is to identify entries where we can predict with high confidence to guarantee a low completion error weighted by evaluated confidence. We provide formulation of the problem, its convex relaxation, and an efficient gradient-based online algorithm due to a formulation of the problem as an online two-player game.

**Randomized rounding and semi-definite relaxations.** For the efficient algorithm, the main analysis technique we deploy is based on randomized rounding solutions of semi-definite programming, due to the seminal work of Goemans and Williamson [1995]. These were originally developed in the context of approximation algorithms for MAX-CUT and other combinatorial problems. Here, we use this analysis technique for analyzing inner products of high dimensional vectors, but in a different context and to argue about a convex relaxation of a continuous optimization problem.

**Abstention in classification and regression.** Abstaining from prediction has a long history in other learning problems such as binary classification Chow [1957]. There are several notable similarities. First, many of the algorithms work as ours does, by first fitting a complete classifier and then deciding afterwards where to abstain [e.g., Hellman, 1970, Goldwasser et al., 2020]. Second, numerous abstention models have been considered, including a fixed cost for abstaining [e.g., Chow, 1957, Kalai and Kanade, 2021], as well as fixing the error rate and maximizing the coverage [e.g., Geifman and El-Yaniv, 2017], as in our work, and fixing the coverage rate while minimizing the error [e.g., Geifman and El-Yaniv, 2019]. Third, there are algorithmic similarities, in particular the observation that worst-case performance guarantees do not depend on knowing the underlying distribution Kalai and Kanade [2021]. Online models of abstention have also been considered [e.g., Li et al., 2011]. For detailed survey on selective classification see [El-Yaniv and Wiener, 2010].

## 2 Problem Setup and Preliminaries

For a natural number $n$, let $[n] = \{1, 2, \ldots, n\}$. Consider a fixed set of indices $\mathcal{X} = [m] \times [n]$ throughout. Thus, in this paper $x \in \mathcal{X}$ is $x = (i, j)$ where $i \in [m], j \in [n]$. The set of $m \times n$ real-valued matrices is thus written as $\mathbb{R}^{\mathcal{X}}$. We will thus view a matrix $M$ over index set $\mathcal{X}$ as a function from $\mathcal{X} \to \mathbb{R}$, and for $x \in \mathcal{X}$, by slight abuse of notation denote by both $M_x$ and $M(x)$ the entry of $M$ at index $x$.

We focus on the squared loss in this paper, though it may be interesting to consider other loss functions in future work. Suppose $\nu$ is a distribution over $\mathcal{X}$, for matrices, $M, M'$, we denote by $\ell(\nu, M, M') \stackrel{\text{def}}{=} \mathbb{E}_{x \sim \nu}[(M(x) - M'(x))^2]$; we will also use the shorthand $\|M - M'\|_\nu^2 \stackrel{\text{def}}{=} \ell(\nu, M, M')$.

When $T \in \mathcal{X}^N$ is a sequence of elements of $\mathcal{X}$ of length $N$, we denote by $\ell(T, M, M') \stackrel{\text{def}}{=} \frac{1}{N} \sum_{x \in T}(M(x) - M'(x))^2$ and the shorthand $\|M - M'\|_T^2 \stackrel{\text{def}}{=} \ell(T, M, M')$.

### 2.1 Matrix Classes, Version Space and Generalization

**Matrix Norms.** Given a matrix $M$, we denote by $\|M\|_{2,\infty}$ the maximum row norm of $M$. We denote by $\|M\|_{\max}$ the *max-norm* of $M$ defined as,

$$\|M\|_{\max} \stackrel{\text{def}}{=} \min_{UV^\top = M} \|U\|_{2,\infty} \cdot \|V\|_{2,\infty}. \tag{2}$$

The trace norm of a matrix $M$, denoted by $\|M\|_{\text{tr}}$, is the sum of its singular values. The Frobenius norm of a matrix $M$, denoted by $\|M\|_{\text{fr}}$, is the square root of the sum of its squared entries.

We will restrict attention to matrices with entries in $[-1, 1]$. The following three classes of matrices with restrictions on respectively the *rank*, *max-norm* and *trace-norm* are of interest in this work. Formally, let $\tilde{\mathcal{K}} = \{M \in [-1, 1]^{\mathcal{X}}\}$, and define,

$$\mathcal{M}_k^{\text{rk}} \stackrel{\text{def}}{=} \{\text{rank}(M) \leq k\} \cap \tilde{\mathcal{K}}, \quad \mathcal{M}_K^{\max} \stackrel{\text{def}}{=} \{\|M\|_{\max} \leq K\} \cap \tilde{\mathcal{K}}, \quad \mathcal{M}_\tau^{\text{tr}} \stackrel{\text{def}}{=} \{\|M\|_{\text{tr}} \leq \tau\} \cap \tilde{\mathcal{K}}.$$

The standard assumption in matrix completion is that the target matrix is low-rank; however, most completion algorithms exploit convex optimization methods and optimize over max-norm or trace-norm bounded matrices. It is known that $\|M\|_{\max} \leq \sqrt{\text{rank}(M)}$ if $M \in [-1, 1]^{\mathcal{X}}$ [cf. Linial et al., 2007, Lemma 4.2] and it is easy to see that $\|M\|_{\text{tr}} \leq \text{rank}(M)\sqrt{mn}$ for $M \in [-1, 1]^{\mathcal{X}}$. Thus, $\mathcal{M}_k^{\text{rk}} \subseteq \mathcal{M}_{\sqrt{k}}^{\max}$ and $\mathcal{M}_k^{\text{rk}} \subseteq \mathcal{M}_{k\sqrt{mn}}^{\text{tr}}$. Hence, we will work with the latter two classes which satisfy the following lemma which is proved in Appendix A.

**Lemma 5.** *The classes, $\mathcal{M}_K^{\max}$ and $\mathcal{M}_\tau^{\text{tr}}$, are closed under negations and are convex.*

**Version Space.** We define the notion of version spaces that are used in our key results. For a sequence $T \in \mathcal{X}^N$, for any class of matrices $\mathcal{M}$, a matrix $M \in \mathcal{M}$ and $\beta > 0$, we define the version space around $M$ of radius $\beta$ based on $T$ w.r.t. $\mathcal{M}$ as

$$\mathcal{V}(M, \beta, T; \mathcal{M}) = \{M' \in \mathcal{M} \mid \ell(T, M, M') \leq \beta\}. \tag{3}$$

Intuitively, version space is the set of matrices in a particular matrix class that are "close" to a given matrix with respect to $T$.

**Generalization Bounds.** For a general class of matrices $\mathcal{M}$, we define a notion of sample complexity that will guarantee the proximity of empirical and population measures of interest for all matrices in the class; we denote this notion by $\mathsf{sc}(\varepsilon, \delta, \mathcal{M})$.

**Definition 6** (Sample Complexity)**.** For $\varepsilon, \delta > 0$ and a class of matrices $\mathcal{M}$, denote by $\mathsf{sc}(\varepsilon, \delta, \mathcal{M})$, the sample complexity, to be the smallest natural number $N_0$, such that for any distribution $\mu$ over $\mathcal{X}$, for $S \sim \mu^{N_0}$, and for any fixed $\hat{M} \in \mathcal{M}$ (possibly depending on $S$), with probability at least $1 - \delta$,

$$\sup_{M \in \mathcal{M}} \left| \ell(S, M, \hat{M}) - \ell(\mu, M, \hat{M}) \right| \leq \varepsilon.$$

If no such $N_0$ exists, $\mathsf{sc}(\varepsilon, \delta, \mathcal{F}) \stackrel{\text{def}}{=} \infty$.

Bounds on the sample complexity for matrix completion can be derived in terms of rank, max-norm and trace-norm using standard results in the literature, and the fact that the squared loss is 2-Lipschitz and bounded by 4 when both its arguments take values in $[-1, 1]$. In the proposition below, the max-norm result follows from Theorem 5 in [Srebro and Shraibman, 2005] and the trace-norm result follows from Theorem 4 in [Shamir and Shalev-Shwartz, 2014].

**Proposition 7.** *For the classes, $\mathcal{M}_K^{\max}$, $\mathcal{M}_\tau^{\mathrm{tr}}$, the following hold,*

1. $\mathsf{sc}(\varepsilon, \delta, \mathcal{M}_K^{\max}) = O\left( \frac{1}{\varepsilon^2} \left( K^2(m+n) + \log \frac{1}{\delta} \right) \right).$

2. $\mathsf{sc}(\varepsilon, \delta, \mathcal{M}_\tau^{\mathrm{tr}}) = O\left( \frac{1}{\varepsilon^2} \left( \tau \sqrt{m+n} + \log \frac{1}{\delta} \right) \right).$

It is worth comparing the two bounds in Proposition 7 above. Consider the matrix $M^\star$ consisting of all 1's – from a matrix completion point of view this is particularly easy as every "user" likes every "movie". Note however that $\|M^\star\|_{\max} = 1$ and $\|M^\star\|_{\mathrm{tr}} = \sqrt{mn}$. For this example, ignoring the dependence on $\varepsilon, \delta$, the sample complexity bound obtained using the trace-norm result would be $O((m+n)\sqrt{m+n})$, while that using max-norm would be $O(m+n)$. In general, it is always the case that $\|M\|_{\mathrm{tr}}/\sqrt{mn} \leq \|M\|_{\max}$, so it may seem that the bounds in terms of trace-norm are weaker. However, there are matrices for which this gap can be large and the sample complexity bound in terms of trace-norm is shown to be tight in general (see [Srebro and Shraibman, 2005, Shamir and Shalev-Shwartz, 2014] for further details).

## 2.2 Full Completion Problem (with Noise)

We now define the matrix completion problem with zero-mean noise, in fact a matrix *estimation* problem in the noisy case. Let $\mathcal{D}$ be a distribution supported on $\mathcal{X} \times [-1, 1]$; the results in the paper can all be easily extended when $\mathcal{D}$ is supported on $\mathcal{X} \times [-B, B]$, increasing squared error by a $B^2$ factor. Let $\mu$ be the marginal distribution of $\mathcal{D}$ over $\mathcal{X}$. Let $S_{XY} = \langle (x_t, y_t) \rangle_{t=1}^N$ be an iid sample drawn from $\mathcal{D}^N$. Note here that $x_t = (i_t, j_t)$ denotes the index of the matrix and $y_t$ the observed value. We let $S = (x_1, \dots, x_N)$ denote the sequence of $x_t$'s from $S_{XY}$ (with repetitions allowed); note that $S$ is distributed as $\mu^N$. Let $M^\star \in [-1, 1]^{\mathcal{X}}$ be the matrix where $M_{ij}^\star = \mathbb{E}_{\mathcal{D}}[Y | X = (i, j)]$. We say that $\hat{M}$ is an $\varepsilon$-accurate completion of $M^\star$, if $\ell(\mu, M^\star, \hat{M}) \leq \varepsilon$.

**Definition 8** (Full Completion Algorithm)**.** We say that $\mathsf{FullComp}(S_{XY}, \varepsilon, \delta, \mathcal{M})$ is a full completion algorithm with sample complexity $\mathsf{sc}_{\mathsf{FC}}(\varepsilon, \delta, \mathcal{M})$ for $\mathcal{M}$, if provided $S_{XY} \sim \mathcal{D}^N$ for some $\mathcal{D}$ over $\mathcal{X} \times [-1, 1]$ with $M^\star \in \mathcal{M}$, $N \geq \mathsf{sc}_{\mathsf{FC}}(\varepsilon, \delta, \mathcal{M})$, $\mathsf{FullComp}(S_{XY}, \varepsilon, \delta, \mathcal{M})$ outputs $\hat{M} \in \mathcal{M}$ that with probability at least $1 - \delta$ satisfies, $\ell(\mu, M^\star, \hat{M}) \leq \varepsilon$.

The following result follows from [Srebro and Shraibman, 2005, Shamir and Shalev-Shwartz, 2014] (see also Prop. 7).

**Proposition 9.** *There exists polynomial time (in $mn/\varepsilon$) full completion algorithms for $\mathcal{M}_K^{\max}$ and $\mathcal{M}_\tau^{\mathrm{tr}}$ (e.g. ERM methods) with the following sample complexity bounds:*

1. $\mathsf{sc}_{\mathsf{FC}}(\varepsilon, \delta, \mathcal{M}_K^{\max}) = O\left( \frac{1}{\varepsilon^2} \left( K^2(m+n) + \log \frac{1}{\delta} \right) \right).$

2. $\mathsf{sc}_{\mathcal{M}_\tau^{\mathrm{tr}}}(\varepsilon, \delta, \mathcal{M}_\tau^{\mathrm{tr}}) = O\left( \frac{1}{\varepsilon^2} \left( \tau \sqrt{m+n} + \log \frac{1}{\delta} \right) \right).$

## 2.3 The Partial Matrix Completion Problem

Recall the definition of the (fractional) partial matrix completion in Eq. 1 and Def. 1. In this work, we will assume that a full completion matrix $\hat{M}$ is already obtained. The focus of the present work

is finding a completion matrix $C \in \{0,1\}^{\mathcal{X}}$ (or $C \in [0,1]^{\mathcal{X}}$ for fractional coverage) with large *coverage*. The coverage in both cases are defined as: $|C| \stackrel{\text{def}}{=} \sum_{x \in \mathcal{X}} C_x$, and a low *loss*, defined as:

$$\ell(C, M^\star, \hat{M}) \stackrel{\text{def}}{=} \frac{1}{|C|} \sum_{x \in \mathcal{X}} C_x (M_x^\star - \hat{M}_x)^2.$$

Note that $C \in [0,1]^{\mathcal{X}}$ can be viewed as a "fractional" set and we are overloading the notation $\ell$ to allow that. Such a fractional coverage can be randomly rounded to a set whose size is within 1 of $|C|$. In an ideal world, we would have at most $\varepsilon$ loss measured over the cells we complete as defined above, and large or even full coverage $|C| = mn$.

Note that although the requirement in the optimization problems in Eq. (1) and Def. 1 is to guarantee $\ell(C, M^\star, M) \leq \varepsilon$, as we don't know $M^\star$, the only guarantee we have by using a full completion algorithm (and generalization bounds) is that (with high probability) $M^\star$ is in some version space, say $\mathcal{V}$, centered at $\hat{M}$. So we actually show a stronger guarantee that, $\sup_{M \in \mathcal{V}} \ell(C, M, \hat{M}) \leq \varepsilon$.

A second observation is that if $M$ is a version space around $\hat{M}$, then by convexity of the matrix classes, the matrix $(M - \hat{M})/2$, must be in some version space, say $\mathcal{V}_0$, centered around the zero matrix, $\mathbf{0}_{\mathcal{X}}$. Thus, we will actually find a $C$ that guarantees, $\ell(C, M, \mathbf{0}_{\mathcal{X}}) \leq \varepsilon$ for every $M \in \mathcal{V}_0$. This means that for maximizing coverage our algorithm has the remarkable property we only need to know the locations of the revealed entries as indicated by $S$.

Section 3 presents a computationally inefficient but statistically superior algorithm in terms of sample complexity for the *coverage problem* and its consequences for partial matrix completion. Section 4 presents a computationally efficient algorithm at the cost of a slightly worse sample complexity when using the class $\mathcal{M}_K^{\max}$.

# 3 An Inefficient Algorithm

The main novelty in the partial matrix completion problem is that of finding an optimal coverage, defined as the matrix $C$ in the formulation in Def. 1. In this section we give an (inefficient) algorithm for finding the optimal coverage, and the generalization error bounds for Partial Matrix Completion that arise from it. In the next section we give an efficient approximation algorithm for doing the same, with slightly worse sample complexity bounds and more complex analysis. This differs substantially from prior work on abstention in classification and regression [Kalai and Kanade, 2021] where the optimal solution can be found in polynomial time.

Let $C \in [0,1]^{\mathcal{X}}$ be a target confidence matrix. For such a $C$, we will denote by $\nu_C$ the probability distribution where $\nu_C(x) = C_x / \|C\|_1$ for $x \in \mathcal{X}$. Let $S \sim \mu^N$ be a sample obtained from the target distribution $\mu$. We output a $C$ which is an optimal solution to the following optimization problem, inspired by a similar problem studied by Kalai and Kanade [2021] for classification with abstention.

$$
\begin{aligned}
&\text{Parameters and inputs}: \gamma, \beta, S, \mathcal{M} \\
&\text{maximize} \quad \|C\|_1 \\
&\text{subject to } C \in [0,1]^{\mathcal{X}}, \text{ and } \quad \forall M \in \mathcal{V}(\mathbf{0}_{\mathcal{X}}, \beta, S; \mathcal{M}), \ \ell(\nu_C, M, \mathbf{0}_{\mathcal{X}}) \leq \gamma
\end{aligned}
\tag{4}
$$

---

**Algorithm 1**

---

1: **Inputs**: $S_{XY} = \langle (x_t, y_t) \rangle_{t=1}^N \sim \mathcal{D}^N$, $\varepsilon$, $\delta$, $\mathcal{M}$, FullComp (cf. Def. 8).
2: Obtain $\hat{M} \in \mathcal{M}$ using FullComp$(S_{XY}, \varepsilon/4, \delta/3, \mathcal{M})$.
3: Obtain $C$ using MP 4 with $\gamma \stackrel{\text{def}}{=} \varepsilon/4$, $\beta \stackrel{\text{def}}{=} \varepsilon/8$, $S = (x_1, x_2, \ldots, x_N)$ from $S_{XY}$.
4: **return** $(\hat{M}, C)$.

---

MP 4 defines a family of optimization problems; we will typically use $\mathcal{M}$ to be $\mathcal{M}_K^{\max}$ or $\mathcal{M}_\tau^{\text{tr}}$. The above optimization problem is in fact a linear program. The only unknowns are $C_x$ for $x \in \mathcal{X}$. However, the set of constraints is infinite and it is unclear how a separation oracle for the constraint set

may be designed. Such an oracle could be designed by solving the following optimization problem:

$$\max_{M \in \mathcal{V}(\mathbf{0}_{\mathcal{X}}, \beta, S; \mathcal{M})} \ell(\nu_C, M, \mathbf{0}_{\mathcal{X}}). \tag{5}$$

This optimization problem requires maximizing a quadratic function, and we prove it to be NP-hard with the additional PSD and symmetric constraints in Appendix A.2.

The following result subsumes Corollary 2. We defer the proof of this theorem to Appendix A, as the proof is similar to that of our efficient algorithm in the next section.

**Theorem 10.** *Let $\mathcal{M}$ be either $\mathcal{M}_K^{\max}$ or $\mathcal{M}_\tau^{\mathrm{tr}}$. Let $\mathcal{D}$ be distribution over $\mathcal{X} \times [-1, 1]$, $\mu$ the marginal of $\mathcal{D}$ over $\mathcal{X}$, $\mu_{\max} = \max_{i,j} \mathbb{P}_\mu((i,j) \text{ is sampled})$. Suppose that $M^\star$, defined as $M_{ij}^\star = \mathbb{E}_{(X,Y) \sim \mathcal{D}}[Y | X = (i,j)]$ satisfies that $M^\star \in \mathcal{M}$. Furthermore, suppose that $S_{XY} \sim \mathcal{D}^N$ and that $\mathsf{FullComp}$ is a full completion algorithm as in Defn. 8. Then, provided $N \geq \max\{\mathsf{sc}_{\mathrm{FC}}(\varepsilon/4, \delta/3, \mathcal{M}), \mathsf{sc}(\varepsilon/8, \delta/3, \mathcal{M})\}$, for $(\hat{M}, C)$ output by Alg. 1, it holds that:*

1. *$\|C\|_1 \geq 1/\mu_{\max}$,*

2. *$\dfrac{1}{\|C\|_1} \sum_{x \in \mathcal{X}} C_x (\hat{M}_x - M_x^\star)^2 \leq \varepsilon$.*

## 4 An Efficient Algorithm

In this section, we show how the result from Proposition 14 can be achieved using an efficient algorithm at a modest cost in terms of sample complexity when using the matrix class $\mathcal{M}_K^{\max}$. In particular, we consider the optimization problem defined in MP 6, where the constraint $\|M\|_{\nu_C}^2 \leq \gamma$ for all $M \in \mathcal{V}(\mathbf{0}_{\mathcal{X}}, \beta, S; \mathcal{M}_K^{\max})$ from MP 4 is replaced by $\mathbb{E}_{x \sim \nu_C}[M_x] \leq \gamma$. The efficiency comes from the fact that we can now implement a separation oracle for the constraint set by solving, for a given $C$, the problem,

$$\max_{M \in \mathcal{V}(\mathbf{0}_{\mathcal{X}}, \beta, S, \mathcal{M}_K^{\max})} \mathbb{E}_{x \sim \nu_C}[M_x].$$

Mathematical program 6 is a convex optimization problem since the constraint set is convex and the objective function is linear. However, it is not immediately clear how this relaxed optimization problem relates to the original, which is the main technical contribution of this section.

---

Parameters and inputs : $\gamma, \beta, S, \mathcal{M}_K^{\max}$

maximize $\quad \|C\|_1$ $\hspace{4cm}$ (6)

subject to $C \in [0,1]^{\mathcal{X}}$, and $\forall M \in \mathcal{V}(\mathbf{0}_{\mathcal{X}}, \beta, S; \mathcal{M}_K^{\max})$, $\mathbb{E}_{x \sim \nu_C}[M_x] \leq \gamma$

---

**Algorithm 2** (Efficient) Offline Algorithm for Partial Matrix Completion

1: **Inputs:** $S_{XY} = \langle (x_t, y_t) \rangle_{t=1}^N \sim \mathcal{D}^N$, $\mathsf{FullComp}$ (cf. Def. 8), $\varepsilon, \delta$.
2: Obtain $\hat{M} \in \mathcal{M}_K^{\max}$ using $\mathsf{FullComp}(S_{XY}, \varepsilon^2/(4\pi^2 K^2), \delta/3, \mathcal{M}_K^{\max})$.
3: Obtain $C$ using MP 6 with $\gamma \overset{\mathrm{def}}{=} \varepsilon/(2\pi K)$, $\beta \overset{\mathrm{def}}{=} \varepsilon^2/(8\pi^2 K^2)$, $S = (x_1, x_2, \ldots, x_N)$ from $S_{XY}$.
4: **return** $(\hat{M}, C)$.

---

The following result subsumes Corollary 3.

**Theorem 11.** *Let $\mathcal{D}$ be a distribution over $\mathcal{X} \times [-1, 1]$, $\mu$ the marginal of $\mathcal{D}$ over $\mathcal{X}$, $\mu_{\max} = \max_{i,j} \mathbb{P}_\mu((i,j) \text{ is sampled})$. Suppose that $M^\star$, defined as $M_{ij}^\star = \mathbb{E}_{(X,Y) \sim \mathcal{D}}[Y | X = (i,j)]$ satisfies that $M^\star \in \mathcal{M}_K^{\max}$. Furthermore, suppose that $S_{XY} \sim \mathcal{D}^N$ and that $\mathsf{FullComp}$ is a full completion algorithm as in Defn. 8. Then, provided $N \geq \max\{\mathsf{sc}_{\mathrm{FC}}(\varepsilon^2/(4\pi^2 K^2), \delta/3, \mathcal{M}_K^{\max}), \mathsf{sc}(\varepsilon^2/(8\pi^2 K^2), \delta/3, \mathcal{M}_K^{\max})\}$, for $(\hat{M}, C)$ output by Alg. 2, it holds that:*

1. *$\|C\|_1 \geq 1/\mu_{\max}$,*

2. *$\dfrac{1}{\|C\|_1} \sum_{x \in \mathcal{X}} C_x (\hat{M}_x - M_x^\star)^2 \leq \varepsilon$.*

# 5 Online Setting

The previous section introduces the *offline* setting for the partial matrix completion problem. In Appendix B, we describe the online version of the problem, which is motivated by two important considerations. First, in many applications, the observation pattern is more general than a fixed distribution. It can be a changing distribution or be comprised of adversarial observations. Second, our online algorithm incrementally updates the solution via iterative gradient methods, which is more efficient than the offline methods. For space considerations, details are deferred to Appendix B which contains the setting, definitions, algorithm specification, and main results, and Appendix C which details the proofs. In particular, the online algorithm, called Online Dual Descent (ODD), is described in Algorithm 3, and its regret guarantee in Theorem 18. The online regret guarantee implies the statistical learning guarantees of the previous sections when the support size is a constant fraction of the full matrix, and this implication is spelled out precisely in Corollary 22.

# 6 Experiments and Implementation

The MovieLens dataset ([Harper and Konstan, 2015]) consists of, among other data, a set of users with their rankings of a set of movies. It is a common benchmark for matrix completion because some of the rankings are missing, and one can make predictions on the rankings of user preferences.

We used the dataset differently, aiming to test our online algorithm on generating a satisfying confidence matrix. The experimental procedure is outlined as follows: we used training data from 250 users and their ratings on 250 movies, giving us a total of 5189 completed samples from the incomplete matrix of size $250 \times 250$. We ran our algorithm, ODD (Algorithm 3), to get a confidence matrix $C$. In parallel, we used another standard matrix completion tool, fancyimpute ([Rubinsteyn and Feldman, 2016]), to fill in the missing entries of the matrix and obtain a completion $\hat{M}_f$. After $C$ and $\hat{M}_f$ are obtained, we reveal the true ratings at the missing entries using the validation set and computed the mean squared error of the predicted rating and true rating at each entry. The following plots show the distribution of the mean squared error with respect to the confidence score $C$ assign at the particular entry.

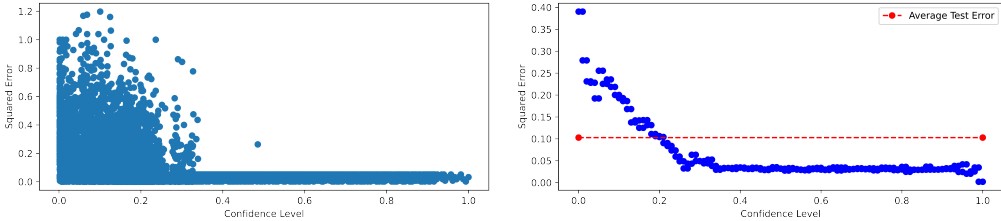

Figure 2: (a) ODD heuristic on $250 \times 250$ user-movie rating data from the MovieLens Dataset. Squared error in entry-wise prediction and entry-wise confidence level estimated by the ODD heuristic. (b) The average squared error over various confidence levels suggests that a higher confidence level is correlated with a lower squared error.

# 7 Conclusion

In this work we define the setting of partial matrix completion, with our high-level contributions outlined as the following:

**A new framework.** We propose a new framework called *partial matrix completion*. In this problem, one is given possibly noisy observations from a low-rank matrix, and the goal is to identify entries that can be predicted with high confidence. The partial matrix completion problem answers two key questions:

- Based on possibly noisy samples from an arbitrary unknown sampling distribution $\mu$, *how many entries can be completed* with $\leq \varepsilon$ error?
- How can we efficiently identify the entries that can be accurately completed?

When the underlying matrix has low rank $k$, we show that it is possible to complete a large fraction of the matrix using only $\tilde{O}(k(m + n))$ observations while simultaneously guaranteeing high accuracy over the completed set. We then study the complexity of identifying the optimal completion matrix. We show that a naïve mathematical programming formulation of the problem is hard. However, we propose a relaxation that gives rise to efficient algorithms, which results in a slightly worse dependence on $k$ for the sample complexity. These guarantees are outlined both in Corollary 2 and Corollary 3, and Theorem 10 and Theorem 11 for their more general versions.

**Online game formulation.** Furthermore, we consider the partial matrix completion problem in the online setting, where the revealed observations are *not* required to follow any particular fixed distribution. The goal henceforth is to minimize *regret*, the gap between the algorithm's performance and the single best decision in hindsight. This is a more general setting, as when imposed with distribution assumptions, regret guarantees in online learning algorithms naturally translate to statistical guarantees.

Our proposed online partial matrix completion algorithm is derived from an online repeated game. The version space is the set of all valid completions with low generalization error. High confidence should be assigned to entries where all completions in the version space are similar. Therefore, we formulate the problem as a two-player online game. One player iteratively updates a confidence matrix, and the other learns the version space. We gave an iterative gradient-based method with provable regret guarantees and concluded with preliminary experimental evidence of the validity of our framework.

## 8 Acknowledgements

Elad Hazan acknowledges funding from the Office of Naval Research grant N000142312156, the NSF award 2134040, and Open Philanthropy. This work was done in part when Clara Mohri was visiting and supported by Princeton University.

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
