# Contents

# A  Supporting Proofs

## A.1  Proof of Lemma 5

**Lemma 5.** *The classes, $\mathcal{M}_K^{\max}$ and $\mathcal{M}_\tau^{\mathrm{tr}}$, are closed under negations and are convex.*

*Proof of Lemma 5.* Suppose $M$ is a matrix then we can write $M = UV^\top$ such that,

- If $M \in \mathcal{M}_K^{\max}$, $\|U\|_{2,\infty} \cdot \|V\|_{2,\infty} \leq K$.
- If $M \in \mathcal{M}_\tau^{\mathrm{tr}}$, $\|U\|_{\mathrm{fr}} \cdot \|V\|_{\mathrm{fr}} \leq \tau$.

Note that $-M = -UV^\top$, negating the sign of $U$ doesn't affect its norm, so clearly the classes are closed under negation.

The convexity of the class follows directly from that $\|\cdot\|_{\max}$ and $\|\cdot\|_{\mathrm{tr}}$ are well-defined norms.

$\square$

## A.2  Computational Hardness of the Coverage Problem

In this section we show that the optimization problem MP 4 is NP-hard in full generality. The first fact we use is the polynomial time equivalence between linear optimization and separation that was established in the work of Grötschel et al. [1981]. It thus remains to prove NP-hardness of the separation problem, which is given in equation 5, and can be restated as the following mathematical program when our version space is restricted with a trace norm constraint:

$$\max_X \sum_{ij} C_{ij} X_{ij}^2 \qquad (7)$$

$$|X_{ij}| \leq 1, \|X\|_{\mathrm{tr}} \leq k.$$

We give evidence of NP-hardness for this program by considering the same mathematical program added a symmetric positive-definite constraint, as follows.

**Lemma 12.** *Mathematical program (8) is NP-hard to compute, or approximate with factor $k^{1-\varepsilon}$ for any $\varepsilon > 0$.*

$$\max_X \sum_{ij} C_{ij} X_{ij}^2 \qquad (8)$$

$$0 \leq X_{ij} \leq 1, X \succeq 0, X \in \mathrm{Sym}(n),\ \mathrm{tr}(X) \leq k.$$

*Proof.* The proof relies on strong hardness of approximation results for the MAX-CLIQUE problem that were proven in Hastad [1996] and subsequent work.

We prove by reduction from $k$-CLIQUE. Let $G(V, E)$ be an instance of the $k$-CLIQUE problem.

**The reduction.** Given a graph $G$, let

$$\mathbb{R}^{V \times V} \ni C_{ij} = \begin{cases} 1 & (i,j) \in E \ \text{ or } \ i = j \\ 0 & \text{otherwise} \end{cases}.$$

$\square$

We now claim the following:

**Lemma 13.** *The value of mathematical program (7) is at least $k^2$ if $G$ contains clique of size $k$. Conversely, if the value of (7) is at least $k^2$, then $G$ contains a clique of size at least $k$.*

*Proof.* **Completeness** If $G$ has a $k$ clique, then consider the following solution $X$. Let

$$v_i = \begin{cases} 1 & i \in \text{clique} \\ 0 & \text{o/w} \end{cases}$$

and let $X = vv^\top$. Then we have that $\mathrm{tr}X = k$, $X_{ij} \in \{0,1\}$. In addition, by definition of $X$, we have that

$$\sum_{ij} C_{ij} X_{ij}^2 = \sum_{ij} C_{ij} v_i^2 v_j^2 = k^2.$$

**Soundness** Suppose program (8) has a solution $X$ with value $k^2$ and trace exactly $k$. The trace equality is w.l.o.g since we can always increase the diagonal entries while preserving constraints and only increasing the objective. Therefore, without loss of generality, we may assume that $\mathrm{tr}(X) = k$.

Next, we claim that w.l.o.g. we have that $\mathrm{rank}(X) = 1$. To see this, notice that we can define a vector $u_i = \sqrt{X_{ii}}$, and $\tilde{X} = uu^\top$. Now, we have that $\mathrm{tr}(\tilde{X}) = k$, and it satisfies the bounded-ness constraints by definition. In addition, we have that the objective is only increased, since

$$\sum_{ij} C_{ij} (v_i v_j)^2 = \sum_{ij} C_{ij} X_{ii} X_{jj} \geq \sum_{ij} C_{ij} X_{ij}^2,$$

where the last inequality is by positive semi-definiteness.

Thus, we can restrict our attention to the set of solutions given by $\mathcal{K}_k = \{uu^\top, \; 0 \leq u_i \leq 1, \; \|u\|_1 = k\}$. We claim that $\mathcal{K}_k$ can be alternatively characterized as the convex hull of all rank-one matrices of the following form:

$$\mathcal{K}_k = \mathrm{conv}\left\{ vv^\top | v \in \{0,1\}^V, \; \|v\|_1 = k \right\}.$$

This fact is shown in page 279 of Warmuth [2010].

We can now continue with the soundness proof. Since the objective $\sum_{ij} C_{ij} X_{ij}^2$ is a convex function, given a distribution over points in $\mathcal{K}_k$, the maximum is obtained in a vertex. Thus, there exists a binary vector $v$ such that its trace is $k$, and for which $\sum_{ij} C_{ij} v_i v_j = k^2$.

Define a subgraph according to $v$ in the natural way: $i \in S$ if and only if $v_i = 1$. Notice that the subset of vertices $S$ is of size $k$ due to the trace.

In terms of number of edges in this subgraph, notice that

$$|E(S)| \quad = \tfrac{1}{2} \sum_{ij \in S} \mathbf{1}_{(i,j) \in E} = \tfrac{1}{2} \sum_{ij} C_{ij} v_i v_j = \tfrac{k^2}{2}.$$

Thus, we have found a clique of size $k$. $\qquad\square$

We note that, although we have shown NP-hardness for the trace-norm with symmetric PSD constraints, it is possible that the optimization problem is efficiently solvable for the max-norm or rank.

### A.3 Proof of Theorem 10

Recall our main theorem for the inefficient algorithm, which we prove in this appendix.

**Theorem 10.** Let $\mathcal{M}$ be either $\mathcal{M}_K^{\max}$ or $\mathcal{M}_\tau^{\mathrm{tr}}$. Let $\mathcal{D}$ be distribution over $\mathcal{X} \times [-1,1]$, $\mu$ the marginal of $\mathcal{D}$ over $\mathcal{X}$. Suppose that $M^\star$, defined as $M_{ij}^\star = \mathbb{E}_D[Y|X = (i,j)]$ satisfies that $M^\star \in \mathcal{M}$. Furthermore, suppose that $S \sim \mathcal{D}^N$ and that FullComp is a full completion algorithm as in Defn. 8. Then, provided $N \geq \max\{\mathsf{sc}_{\mathrm{FC}}(\varepsilon/4, \delta/3, \mathcal{M}), \mathsf{sc}(\varepsilon/8, \delta/3, \mathcal{M})\}$, for $(\hat{M}, C)$ output by Alg. 1, with probability at least $1 - \delta$, it holds that:

1. $\|C\|_1 \geq 1/\mu_{\max}$ and

2. $\dfrac{1}{\|C\|_1} \sum_{x \in \mathcal{X}} C_x (\hat{M}_x - M_x^\star)^2 \leq \varepsilon.$

The proof relies on the following proposition.

**Proposition 14.** *Suppose $S \sim \mu^N$ for some distribution $\mu$ over $\mathcal{X}$; let $\mu_{\max} \stackrel{\text{def}}{=} \max_{x \in \mathcal{X}} \mu_x$. Suppose $\mathcal{M}$ is one of $\mathcal{M}_K^{\max}$ or $\mathcal{M}_\tau^{\mathrm{tr}}$, let $N \geq \mathsf{sc}(\gamma/2, \delta, \mathcal{M})$ (cf. Defn. 6) and suppose $C$ is the maximizer of MP 4 with $\beta = \gamma/2$, then with probability at least $1 - \delta$, we have $\|C\|_1 \geq 1/\mu_{\max}$.*

*Proof.* Based on the condition on $N$ and using $\hat{M} = \mathbf{0}_\mathcal{X}$ in Defn. 6, with probability at least $1 - \delta$, the following holds for all $M \in \mathcal{M}$:

$$\left| \|M\|_\mu^2 - \|M\|_S^2 \right| \leq \gamma/2 \tag{9}$$

We need to show that some $C^\mu$, with $\|C^\mu\|_1 \geq 1/\mu_{\max}$ is feasible. Let $C_x^\mu = \mu(x)/\mu_{\max}$. Then, clearly $C_x \in [0,1]$, $\|C^\mu\|_1 = 1/\mu_{\max}$ and $\nu_{C^\mu} = \mu$. Using Eq. (9), we have that, $\|M\|_\mu^2 \leq \|M\|_S^2 + \gamma/2$, and as a result for every $M \in \mathcal{V}(\mathbf{0}_\mathcal{X}, \beta, S; \mathcal{M})$, it holds that $\|M\|_\mu^2 \leq \beta + \gamma/2 = \gamma$. Thus, $C_\mu$ is a feasible solution to the optimization problem with objective value at least $1/\mu_{\max}$. $\square$

*Proof of Theorem 10.* There are three bad events, for each of which we bound their probability by at most $\delta/3$. First, FullComp succeeds with probability at least $1 - \delta/3$.

Second, provided $N \geq \mathsf{sc}(\varepsilon/8, \delta/3, \mathcal{M})$, for $\hat{M}$ output by FullComp, using Defn. 6 it must hold for each $M \in \mathcal{M}$ that with probability at least $1 - \delta/3$,

$$\left| \|M - \hat{M}\|_\mu^2 - \|M - \hat{M}\|_S^2 \right| \leq \varepsilon/8 \tag{10}$$

In particular, when $\beta = \varepsilon/8$, this means that (assuming FullComp has not failed),

$$\|(M^\star - \hat{M})/2\|_S^2 \leq \frac{1}{4}\|M^\star - M\|_\mu^2 + \frac{\epsilon}{32} \leq \frac{\varepsilon}{16} + \frac{\varepsilon}{32} \leq \beta.$$

As a result, and also using the convexity of $\mathcal{M}$ (cf. Lemma 5), $(M^\star - \hat{M})/2 \in \mathcal{V}(\mathbf{0}_\mathcal{X}, \beta, S; \mathcal{M})$.

Finally, for $N \geq \mathsf{sc}(\varepsilon/8 = \gamma/2, \delta/3, \mathcal{M})$, Proposition 14 guarantees with probability at least $1 - \delta/3$, that the output of MP 4 satisfies $\|C\|_1 \geq 1/\mu_{\max}$.

The constraint of MP 4 together with the fact that $(M^\star - \hat{M})/2 \in \mathcal{V}(\mathbf{0}_\mathcal{X}, \beta, S; \mathcal{M})$ guarantees that, $\|M^\star - \hat{M}\|_{\nu_C}^2 \leq 4\gamma$ which completes the required proof. $\square$

### A.4 Proof of Theorem 11

**Proposition 15.** *Suppose $S \sim \mu^N$ for some distribution $\mu$ over $\mathcal{X}$; let $\mu_{\max} \overset{def}{=} \max_{x \in \mathcal{X}} \mu_x$. Let $N \geq \mathsf{sc}(\gamma^2/2, \delta, \mathcal{M}_K^{\max})$ (cf. Defn. 6) and suppose $C$ is the maximizer of MP 6 with $\beta = \gamma^2/2$, then with probability at least $1 - \delta$, we have $\|C\|_1 \geq 1/\mu_{\max}$.*

*Proof of Proposition 15.* Based on the condition on $N$ and using $\hat{M} = \mathbf{0}_\mathcal{X}$ in Defn. 6 with probability at least $1 - \delta$ it holds for every $M \in \mathcal{M}_K^{\max}$:

$$\left| \|M\|_\mu^2 - \|M\|_S^2 \right| \leq \gamma^2/2. \tag{11}$$

As in the proof of Proposition 14, let $C_x^\mu = \mu_x/\mu_{\max}$ and notice that $C_x^\mu \in [0,1]$, $\|C^\mu\|_1 = 1/\mu_{\max}$ and $\nu_{C^\mu} = \mu$. Using Eq. (11), for any $M \in \mathcal{V}(\mathbf{0}_\mathcal{X}, \beta, S; \mathcal{M}_K^{\max})$, we have that $\|M\|_\mu^2 \leq \beta + \gamma^2/2 \leq \gamma^2$. Finally, using the fact that $\mathbb{E}_{x \sim \mu}[M_x] \leq \sqrt{\|M\|_\mu^2}$, we get that $C^\mu$ is feasible and the result follows. $\square$

The key to proving Theorem 11 is the following proposition, whose proof contains the main technical innovation. The derivation of the main theorem given the proposition is in the Appendix.

**Proposition 16.** *For the class $\mathcal{M}_K^{\max}$, for any distribution $\nu$ over $\mathcal{X}$, $\beta > 0$ and $S \in \mathcal{X}^N$, the following holds: for $\mathcal{V} = \mathcal{V}(\mathbf{0}_\mathcal{X}, \beta, S; \mathcal{M}_K^{\max})$,*

$$\sup_{M \in \mathcal{V}} \ell(\nu, M, \mathbf{0}_\mathcal{X}) \leq \frac{\pi K}{2} \sup_{M \in \mathcal{V}} \mathbb{E}_{x \sim \nu}[M_x].$$

*Proof of Proposition 16.* For succinctness let $\mathcal{V}_0 = \mathcal{V}(\mathbf{0}_\mathcal{X}, \beta; S, \mathcal{M}_K^{\max})$. Let $\nu$ be a an arbitrary distribution over $\mathcal{X}$ and since we know that $\mathcal{V}_0$ is closed and bounded, we know that a matrix $M \in \mathcal{V}_0$ exists which achieves the value of $\sup_{M \in \mathcal{V}_0} \ell(\nu, M, \mathbf{0}_\mathcal{X})$. We will use the probabilistic method to show that there exists $\tilde{M} \in \mathcal{V}_0$ such that $\mathbb{E}_{x \sim \nu}[\tilde{M}_x] \geq \frac{2}{\pi K} \ell(\nu, M, \mathbf{0}_\mathcal{X})$.

Since $M \in \mathcal{M}_K^{\max}$, there exist $U, V$ such that $M = UV^\top$ and that $\|U\|_{2,\infty} \cdot \|V\|_{2,\infty} \le K$. Denote by $u_i, v_j$ respectively the $i$-th and $j$-th rows of $U$ and $V$. Suppose $u_i, v_j \in \mathbb{R}^D$ for some finite $D$, we know that we can always choose $D \le \min\{m, n\}$. Let $w$ be a random vector drawn uniformly from the unit sphere in $\mathbb{R}^D$. For any vector $v \in \mathbb{R}^D$, let $\tilde{v} \overset{\text{def}}{=} \operatorname{sign}(v^\top w)v$ and obtain the matrices $\tilde{U}, \tilde{V}$ from $U$ and $V$ by applying this transformation to all the rows of $U$ and $V$.

We define $\tilde{M} = \tilde{U}\tilde{V}^\top$. Note that as the only difference between $U$ and $\tilde{U}$ (resp. $V$ and $\tilde{V}$) is sign changes for some subset of the rows, we have $\|\tilde{U}\|_{2,\infty} = \|U\|_{2,\infty}$ (resp. $\|\tilde{V}\|_{2,\infty} = \|V\|_{2,\infty}$). Also for any $i, j$, $|u_i^\top v_j| = |\tilde{u}_i^\top \tilde{v}_j|$, thus the entries of $M$ and $\tilde{M}$ may only differ in sign. Thus, $\tilde{M} \in \mathcal{M}_K^{\max}$ irrespective of the random choice of the vector $w$. In order to complete the proof by the probabilistic method, it suffices to show that,

$$\mathbb{E}_w \left[ \mathbb{E}_{x \sim \nu}[\tilde{M}_x] \right] \ge \frac{2}{\pi K} \ell(\nu, M, \mathbf{0}_{\mathcal{X}}). \tag{12}$$

Let $\theta(u, v)$ denote the angle between any two vectors $u, v \in \mathbb{R}^D$. For a vector $w$ drawn at random from the unit sphere it can be verified that $\mathbb{P}[\operatorname{sign}(\tilde{u}^\top \tilde{v}) \ne \operatorname{sign}(u^\top v)] = \frac{\theta(u,v)}{\pi}$. We will show:

$$\mathbb{E}_w[\tilde{u}^\top \tilde{v}] \ge \frac{2(u^\top v)^2}{\pi \|u\| \|v\|}. \tag{13}$$

It suffices to show the above for the case when $u^\top v \ge 0$ as flipping the sign of $v$ affects neither $\tilde{v}$ nor $(u^\top v)^2$. Note that in this case $\theta(u, v) \in [-\pi/2, \pi/2]$. To prove Eq. (13), we use:

**Lemma 17.** *For any $\theta \in [-\pi/2, \pi/2]$, we have that, $1 - 2\theta/\pi \ge 2\cos(\theta)/\pi$.*

*Proof of Lemma 17.* Let $f : [-\pi/2, \pi/2] \to \mathbb{R}$, where $f(\theta) = 1 - 2\theta/\pi - 2\cos\theta/\pi$. Note that since $|\sin\theta| \le 1$ for all $\theta$, $f'(\theta) \le 0$, so $f$ is decreasing and $f(\pi/2) = 0$. $\square$

Now, using the fact that $\cos(\theta(u, v)) = u^\top v/(\|u\| \|v\|)$, we get,

$$\mathbb{E}_w[\tilde{u}^\top \tilde{v}] = u^\top v \cdot \left( 1 - 2\mathbb{P}[\operatorname{sign}(\tilde{u}^\top \tilde{v}) \ne \operatorname{sign}(u^\top v)] \right)$$
$$= u^\top v \cdot \left( 1 - \frac{2\theta(u, v)}{\pi} \right) \ge u^\top v \cdot \frac{2\cos(\theta(u, v))}{\pi} = \frac{2(u^\top v)^2}{\pi \|u\| \|v\|}.$$

In the last step, we used Lemma 17. For any fixed $x = (i, j) \in \mathcal{X}$, we have that,

$$\mathbb{E}_w[\tilde{M}_x] = \mathbb{E}_w[\tilde{u}_i^\top \tilde{v}_i] \ge \frac{2(u_i^\top v)^2}{\pi \|u_i\| \|v_j\|} \ge \frac{2M_x^2}{\pi K}.$$

The last inequality follows from the fact that $\|M\|_{\max} = \|U\|_{2,\infty} \cdot \|V\|_{2,\infty} \le K$ and that $M_x = u_i^\top v_j$. Taking expectation with respect to $\nu$ and applying Fubini's theorem establishes (12). $\square$

*Proof of Theorem 11.* Apart from the use of Proposition 16, the proof is essentially the same as that of Theorem 10. We bound the probability of the three undesirable events by $\delta/3$ each. First, FullComp succeeds with probability at least $1 - \delta/3$.

Second, provided $N \ge \operatorname{sc}(\varepsilon^2/(8\pi^2 K^2), \delta/3, \mathcal{M}_K^{\max})$, for $\hat{M}$ output by FullComp, using Defn. 6 it must hold for each $M \in \mathcal{M}_K^{\max}$ that with probability at least $1 - \delta/3$,

$$\left| \ell(\mu, M, \hat{M}) - \ell(S, M, \hat{M}) \right| \le \frac{\varepsilon^2}{8\pi^2 K^2} \tag{14}$$

In particular, when $\beta = \varepsilon^2/(8\pi^2 K^2)$, this means that (assuming FullComp has not failed),

$$\|(M^\star - \hat{M})/2\|_S^2 \le \frac{1}{4}\|M^\star - M\|_\mu^2 + \frac{\epsilon^2}{32\pi^2 K^2} \le \frac{\varepsilon^2}{16\pi^2 K^2} + \frac{\varepsilon^2}{32\pi^2 K^2} \le \beta.$$

As a result, and using the convexity of $\mathcal{M}_K^{\max}$ (cf. Lemma 5), $(M^\star - \hat{M})/2 \in \mathcal{V}(\mathbf{0}_{\mathcal{X}}, \beta, S; \mathcal{M}_K^{\max})$. For $N \ge \operatorname{sc}(\varepsilon^2/(8\pi^2 K^2) = \gamma^2/2, \delta/3, \mathcal{M}_K^{\max})$, Proposition 15 guarantees with probability at least $1 - \delta/3$, that the output of MP 6 satisfies $\|C\|_1 \ge 1/\mu_{\max}$.

Finally, since $(M^\star - \hat{M})/2 \in \mathcal{V}(\mathbf{0}_\mathcal{X}, \beta, S; \mathcal{M}_K^{\max})$, we have:

$$\ell(\nu_C, M^\star, \hat{M}) \leq \sup_{M \in \mathcal{V}(\mathbf{0}_\mathcal{X}, \beta, S; \mathcal{M}_K^{\max})} \ell(\nu_C, M, \mathbf{0}_\mathcal{X}) \leq \frac{\pi K}{2} \cdot \sup_{M \in \mathcal{V}(\mathbf{0}_\mathcal{X}, \beta, S; \mathcal{M}_K^{\max})} \mathbb{E}_{x \sim \nu_C}[M_x] \leq \frac{\pi K}{2} \cdot \gamma.$$

In the penultimate step we used Proposition 16 and in the last step we used the fact that $C$ is feasible for MP 6. Substituting the value of $\gamma$ completes the proof. $\qquad\square$

# B  Online Partial Matrix Completion

This section uses tools from online convex optimization, in particular its application to games, to design an efficient gradient-based online algorithm with provable regret guarantee to find a near optimal confidence matrix to the partial matrix completion problem.

**Organization.**  The organization of this section is as follows: we begin with a short background description on online convex optimization, its application to matrix completion problems and games in Section B.1, explain the main motivations and intuitions of our online algorithm in Section B.2, introduce the general protocol of online partial matrix completion in Section B.3 and the design of objective functions in Section B.4. We formally give the algorithm specification and the regret guarantee in Section B.5. Finally, we show the implication on statistical guarantee in Section B.6.

## B.1  Preliminaries

**Online convex optimization.**  In online convex optimization, a player iteratively chooses a point $x_t \in \mathcal{K} \subseteq \mathbb{R}^d$ at time step $t$, and receives a convex loss function $h_t$, to which the player incurs loss $h_t(x_t)$. The performance of the player is measured by *regret*, the total excess loss incurred by the player's decisions than the best single decision $x^* \in \mathcal{K}$. Formally, regret is given by the following definition

$$\text{Regret}_T \overset{\text{def}}{=} \sum_{t=1}^{T} h_t(x_t) - \min_{x \in \mathcal{K}} \sum_{t=1}^{T} h_t(x).$$

The goal is to design algorithms that achieve sublinear regret, which means that with time the algorithm's decisions converge in performance to the best single decision in hindsight. For a survey of methods and techniques, see [Hazan et al., 2016].

**Online matrix prediction.**  Online convex optimization has been proved useful in solving matrix prediction problems. In particular, Hazan et al. [2012b] give an efficient first-order online algorithm that iteratively produces matrices $M_t$ of low complexity that for any sequence of adversarially chosen convex, Lipschitz loss functions and indices $(i_t, j_t)$'s, the online matrix prediction algorithm gives a regret bound of $\tilde{\mathcal{O}}(\sqrt{(m+n)T})$. The linear dependence on the matrix dimension $m, n$ translates to the convergence in performance to the best complexity-constrained matrix in hindsight after seeing only square root of the total number of entries.

**Games and regret.**  One important branch of theory developed in online convex optimization is its connection to finding the equilibrium point in two-player games, first discovered by Freund and Schapire [1999]. In this framework, two players iteratively pick decisions $x_t \in \mathcal{X}, y_t \in \mathcal{Y}$ at time step $t$, after which a (possibly adversarially) chosen loss function $h_t(x, y)$ is revealed, where $h_t$ is convex in $x$ and concave in $y$. Player 1 incurs loss $h_t(x_t, y_t)$ and player 2 gains reward $h_t(x_t, y_t)$. The objective for the players is to produce a sequence of decisions $\{x_t\}_{t=1}^{T}, \{y_t\}_{t=1}^{T}$ that converges in performance to the best single $x^*, y^* \in \mathcal{K}$ in hindsight. The regret for the two players are given by

$$\text{Regret}_T(\texttt{player1}) = \sum_{t=1}^{T} h_t(x_t, y_t) - \min_{x \in \mathcal{X}} \sum_{t=1}^{T} h_t(x, y_t),$$

$$\text{Regret}_T(\texttt{player2}) = \max_{y \in \mathcal{Y}} \sum_{t=1}^{T} h_t(x_t, y) - \sum_{t=1}^{T} h_t(x_t, y_t).$$

Under distributional assumption of the function $h_t$'s, namely if $h_t$'s are bilinear, i.i.d. stochastically chosen according to some distribution, then the following holds:

$$\left| \mathbb{E}\left[ \sum_{t=1}^{T} h_t(x_t, y_t) \right] - \min_{x \in \mathcal{X}} \max_{y \in \mathcal{Y}} \mathbb{E}\left[ \sum_{t=1}^{T} h_t(x, y) \right] \right|$$
$$\leq \max\{\text{Regret}_T(\texttt{player1}), \text{Regret}_T(\texttt{player2})\}.$$

For such reasons, we define a notion of game regret, denoted Game-Regret$_T$, to be the maximum of the regret incurred by the two players. Sublinear game regret can be used to compute the game equilibrium [Freund and Schapire, 1999].

## B.2 Motivations

Two important aspects lie at heart of the general motivation for considering partial matrix completion in an online setting. First, in many applications, the observation pattern is more general than a fixed distribution. It can be a changing distribution or be comprised of adversarial observations. Second, our online algorithm incrementally updates the solution via iterative gradient methods, which is more efficient than the offline methods.

### B.2.1 Game theoretic nature of partial matrix completion

With the preliminary introduction on saddle-point problems, we will describe their connection to our problem of interest - finding the optimal confidence matrix $C$ in partial matrix completion problems. Recall that we want to find $C$ that (1) has maximal coverage under the constraint that (2) the deviation between any two possible completions $M^1$ and $M^2$ with respect to $C$ is small.

With this objective in mind, we can think of the following two-player game. Player 1 plays a confidence matrix $C$, player 2 plays a pair of possible matrix completions $(M^1, M^2)$. The goal of player 1 is to (a) maximize the coverage of $C$ and (b) minimize the deviation between $M^1$ and $M^2$ with respect to $C$. The goal of player 2 is to maximize the deviation between $M^1$ and $M^2$ with respect to $C$. The equilibrium point in this problem is exactly given by a confidence matrix that simultaneously has high coverage and with respect to which the deviation between any two possible completions $M^1$ and $M^2$ is small. The game theoretic nature of the partial matrix completion problem leads us to consider designing a provably low-game-regret online two-player algorithm.

### B.2.2 Soft constraints

It is worth noting that this formulation differs from the MP4 and MP6 in the absence of hard constraints. In particular, instead of imposing an $\varepsilon$-margin on the deviation between any two possible completions with respect to the confidence matrix, we formulate this objective in the objective functions for player 1 and 2. This allows us to perform fast gradient-based algorithm, which will be detailed soon in the Algorithm 3. The set of possible matrix completions is formally given by the version space, i.e. the set of matrices of low complexity, with bounded norms, and deviate little from the observed data. To avoid computationally expensive projections, Algorithm 3 further removed the constraint on deviation from observations and replaced with a soft constraint that penalizes the completion's deviation from observations. Similar techniques have been seen in [Mahdavi et al., 2012].

## B.3 Online PMC General Protocol

With the motivations explained, we are ready to introduce the general protocol of online matrix completion and then give the details of our objective functions. In the online partial matrix completion problem, the algorithm acts for two players and follows the following protocol. At time step $t$,

1. Player 1 picks a confidence matrix from a constraint set $C_t \in \mathcal{C}$, where $\mathcal{C}$ is defined below. Player 2 picks two matrices $M_t^1, M_t^2 \in \mathcal{M}_K^{\max}$.

2. An adversary reveals a tuple $(x_t, o_t) \in \mathcal{X} \times [-1, 1]$. Based on this tuple, a function is constructed, denoted $h_t(C, M^1, M^2)$, which is concave in $C$ and convex in $M^1, M^2$.

3. Player 1 receives reward $h_t(C_t, M_t^1, M_t^2)$. Player 2 incurs loss $h_t(C_t, M_t^1, M_t^2)$.

The convex-concave function $h_t$, detailed in Section B.4, measures the coverage of $C$, the deviation between $M^1$ and $M^2$ with respect to $C$, and the deviation of $M^1, M^2$ from the observations.

Note that in previous sections, we took $\mathcal{C} = [0, 1]^{\mathcal{X}}$. In this section we consider a more general case, and consider two choices for $\mathcal{C}$. The first choice is the unit simplex $\Delta_{\mathcal{X}} \subset \mathbb{R}^{\mathcal{X}}$. This is natural as it induces a probability distribution of completion confidence over the entries. The second choice is the scaled hypercube $\mathcal{C} = \left[0, \frac{1}{(m+n)^{3/2}}\right]^{\mathcal{X}}$. The scaling is for the mere purpose for analysis and representation of theorems. In B.6, we will show a reduction from the online result to the offline guarantee.

## B.4 Designing Convex-Concave Objectives

The previous sections motivate the design of the convex-concave function $h_t$, where $\alpha, \theta > 0$ are positive parameters:

$$h_t(C, M^1, M^2) = H(C) - \alpha G(C, M^1, M^2) + \alpha \theta f_t(M^1, M^2),$$

where the three components of $h_t$ are:

1. $H(\cdot)$ is a measure of the effective support size of $C$, which can be taken to be either:

    (a) entropy, i.e. $H(C) = -\sum_{x \in \mathcal{X}} C_x \log(C_x)$ defined over the simplex $\mathcal{C} = \Delta_\mathcal{X}$, or

    (b) $H(C) = \|C\|_1$, defined over the scaled hypercube $\mathcal{C} = \left[0, \frac{1}{(m+n)^{3/2}}\right]^\mathcal{X}$.

2. $G(C, M^1, M^2) \overset{\text{def}}{=} \sum_{x \in \mathcal{X}} C_x(M_x^1 - M_x^2)$, a linear relaxation of $\sum_{x \in \mathcal{X}} C_x(M_x^1 - M_x^2)^2$ (see Theorem 24 and Corollary 25 in Appendix C), which measures the deviation of two completions $M^1, M^2$ with respect to $C$.

3. $f_t(M^1, M^2) \overset{\text{def}}{=} \left[(M_{x_t}^1 - o_t)^2 + (M_{x_t}^2 - o_t)^2\right]$, which measures the deviation of the two completions from observation made at time $t$. This serves as a soft constraint where $M^1, M^2$ minimizing $h_t$ will have values close to $o_t$ at entry $x_t$.

With slight abuse of notation, we denote $M_t \overset{\text{def}}{=} (M_t^1, M_t^2)$ in the following sections for convenience and presentation clarity. Note that $h_t$ is concave in $C$ and convex in $M^1, M^2$. Similar to the previous section, remark that we can, albeit suffering a constant in all performance metrics, consider that all observations $o_t$ are zero and maintain a single matrix in place of $M_t^1, M_t^2$ that measures the radius of the version space.

## B.5 Online Dual Descent (ODD): online gradient-based algorithm for partial matrix completion

Formally, we propose Online Dual Descent (ODD, Algorithm 3):

---

**Algorithm 3** Online Dual Descent for Partial Matrix Completion (ODD)

---

1: **Input**: Gradient-based online coverage and matrix update functions $\mathcal{A}_C, \mathcal{A}_M$, parameters $\eta > 0$, $\alpha, \theta > 0$.
2: Initialize $C_1, M_1 \leftarrow \mathcal{A}_C(\emptyset), \mathcal{A}_M(\emptyset)$.
3: **for** $t = 1, 2, ..., T$ **do**
4:    Player 1 plays $C_t$; player 2 plays $M_t$.
5:    Adversary draws tuple $(x_t, o_t)$ and constructs function $h_t$ with parameters $\alpha, \theta$.
6:    Player 1 receives reward $h_t(C_t, M_t)$, player 2 incurs loss $h_t(C_t, M_t)$.
7:    Player 1 updates $C_{t+1} \leftarrow \mathcal{A}_C(C_t, M_t)$. Player 2 updates $M_{t+1} \leftarrow \mathcal{A}_M(C_t, M_t, (x_t, o_t))$.
8: **end for**
9: **output**:s $\bar{C} = \frac{1}{T} \sum_{t=1}^T C_t$.

---

Here, $\mathcal{A}_C : \mathcal{C} \times (\mathcal{M}_K^{\max})^2 \to \mathcal{C}$ and $\mathcal{A}_M : \mathcal{C} \times (\mathcal{M}_K^{\max})^2 \times (\mathcal{X}, [-1, 1]) \to (\mathcal{M}_K^{\max})^2$. The detailed analysis of the algorithm will be deferred to the following section and Appendix C, but we will first state its regret guarantee:

**Corollary 18.** *For any sequence of $\{(x_t, o_t)\}_{t=1}^T$, Algorithm 3 gives the following regret guarantee on the obtained set $\{(C_t, M_t)\}_{t=1}^T$, such that for settings 1a, 1b,*

$$\textit{Game-Regret}_T \overset{\text{def}}{=} \max\left\{\textit{Regret}_T^{\mathcal{A}_C}, \alpha \cdot \textit{Regret}_T^{\mathcal{A}_M}\right\} \leq \tilde{O}(K\alpha\theta\sqrt{(m+n)T}).$$

We henceforth denote the upper bound on the regret by

$$\text{Game-Regret}_T \leq B_T.$$

The main theorem regarding the game regret of Algorithm 3 is implied by the existence of low regret guarantee for gradient-based subroutines $\mathcal{A}_C, \mathcal{A}_M$, outlined in the following theorems. The details of the subroutines and analysis are deferred to Appendix C.

**Theorem 19.** *Denote at every time step $t$, consider the concave reward function $r_t(C) \stackrel{def}{=} H(C) - \alpha G(C, M_t)$. There exists a sub-routine gradient-based update $\mathcal{A}_C$ (see Alg. 4 for an example) with the following regret guarantee w.r.t. $r_t$:*

1. *For setting 1a,*

$$Regret_T^{\mathcal{A}_C} \stackrel{def}{=} \max_{C \in \mathcal{C}} \sum_{t=1}^{T} r_t(C) - \sum_{t=1}^{T} r_t(C_t) \leq O(\alpha \sqrt{\log(mn)T}) = \tilde{O}(\alpha \sqrt{T}).$$

2. *For setting 1b,*

$$Regret_T^{\mathcal{A}_C} \stackrel{def}{=} \max_{C \in \mathcal{C}} \sum_{t=1}^{T} r_t(C) - \sum_{t=1}^{T} r_t(C_t) \leq O(\alpha \sqrt{(m+n)T}).$$

**Theorem 20** (adapted from Hazan et al. [2012b]). *Denote at every time step $t$, consider the concave reward function $\gamma_t(M) \stackrel{def}{=} G(C_t, M) - \theta f_t(M)$. There exists a sub-routine gradient-based update $\mathcal{A}_M$ (see Alg. 5 for an example) such that, under either setting 1a or 1b, the following regret guarantee w.r.t. $\gamma_t$ holds:*

$$Regret_T^{\mathcal{A}_M} \stackrel{def}{=} \max_{M \in \mathcal{M}_K^{\max} \times \mathcal{M}_K^{\max}} \sum_{t=1}^{T} \gamma_t(M) - \sum_{t=1}^{T} \gamma_t(M_t) \leq O(K\theta \sqrt{(m+n)T}).$$

### B.6 Offline implications

In this section we show that how the regret guarantee we obtained in the online setting translates to an offline performance guarantee on $C$. The offline implications hold under the following assumptions of the revealed entries:

**Assumption 1.** *At each time step $t$, the index $x_t = (i_t, j_t)$ is sampled according to some unknown sampling distribution $\mu$, and the observation $o_t = M^\star(i_t, j_t)$, where $M^\star \in \mathcal{M}_K^{\max}$ is the ground truth matrix.*

Consider the following empirical and general version spaces:

$$\mathcal{V}_{T,\delta} \stackrel{def}{=} \left\{ M \in \mathcal{M}_K^{\max} \mid \frac{1}{T} \sum_{t=1}^{T} (M_{x_t} - o_t)^2 \leq \delta \right\},$$

$$\mathcal{V}_\delta \stackrel{def}{=} \left\{ M \in \mathcal{M}_K^{\max} \mid \mathop{\mathbb{E}}_{(x,o) \sim \mathcal{D}} [(M_x - o)^2] \leq \delta \right\}.$$

**Lemma 21.** *After $T$ iterations, and assume that for some $\delta > 0$,*

$$\frac{1}{\theta} \left( 2D + \frac{Regret_T^{\mathcal{A}_M}}{T} \right) \leq \frac{\delta^{2/3}}{2},$$

*with $D = 1$ in setting 1a, and $D = \sqrt{m+n}$ in setting 1b. The following properties hold on the obtained $\bar{C} \stackrel{def}{=} \frac{1}{T} \sum_{t=1}^{T} C_t$ returned by Algorithm 3: with probability $\geq 1 - \exp\left(-\frac{\delta^{4/3}T}{512}\right)$,*

$$H(\bar{C}) - \max_{M \in \mathcal{V}_{T,\delta}^2} \alpha \cdot G(\bar{C}, M) \geq \max_{C \in \mathcal{C}} \min_{M \in \mathcal{V}_{\delta^{2/3}}^2} \{H(C) - \alpha \cdot G(C, M)\} - 2\alpha\theta\delta - \frac{B_T}{T}.$$

Lemma 21 implies the following guarantee.

**Theorem 22.** *Suppose the underlying sampling distribution is $\mu$. Let $C_\mu \in \mathcal{C}$ be its corresponding confidence matrix. In particular, for setting 1a, $C_\mu = \mu$, i.e. $(C_\mu)_{ij} = \mathbb{P}_\mu((i,j)$ is sampled); for setting 1b, $C_\mu$ satisfies that $C_\mu / \|C_\mu\|_1 = \mu$. Then, for any $\delta > 0$, Algorithm 3 run with $\alpha = O(\delta^{-1/6})$ returns a $\bar{C}$ that guarantees the following bounds: with probability at least $1 - c_1 \exp(-c_2 \delta^2 T)$ for some universal constants $c_1, c_2 > 0$,*

1. *For setting 1a, take $\theta = O(\delta^{-2/3})$, after $T = \tilde{O}(\delta^{-2} K^2(m+n))$ iterations,*

*(a)* $H(\bar{C}) \geq H(C_\mu) - O(\delta^{1/6})$.

*(b)* $\displaystyle\max_{M^1, M^2 \in \mathcal{V}_\delta} G(\bar{C}, M^1, M^2) \leq O(\delta^{1/6} \log(mn))$.

2. *For setting* 1b *take* $\theta = O(\delta^{-2/3}\sqrt{m+n})$, *after* $T = \tilde{O}(\delta^{-2} K^2 (m+n))$ *iterations,*

*(a)* $\|\bar{C}\|_1 \geq \|C_\mu\|_1 - O(\delta^{1/6}\sqrt{m+n})$.

*(b)* $\displaystyle\max_{M^1, M^2 \in \mathcal{V}_\delta} G(\bar{C}, M^1, M^2) \leq O(\delta^{1/4}\sqrt{m+n})$.

**Remark 23.** We explain the implication of the above theorem. Suppose the sampling distribution is supported uniformly across a constant fraction $0 < c \leq 1$ of the entire matrix. This implies that (1) the obtained confidence matrix $\bar{C}$ has a coverage lower bounded by $1 - \delta^{1/6}$ fraction of the true distribution, and (2) $\bar{C}$ induces a weighted maximal distance on the version space $\mathcal{V}_\delta$:

$$\max_{M^1, M^2 \in \mathcal{V}_\delta} \left\{ \frac{1}{\|\bar{C}\|_1} \sum_{i \in [m], j \in [n]} \bar{C}_{ij} (M^1_{ij} - M^2_{ij})^2 \right\} \leq O(\delta^{1/6}).$$

Also see Corollary 4 for this example.

# C  Supporting Proofs for Appendix B

## C.1  Linear Relaxation

This section follows similarly to that of Proposition 16.

**Theorem 24.** *Consider the following two functions defined on $\mathcal{C}$:*

$$\tilde{g}(C) \stackrel{def}{=} \sup_{M \in \mathcal{V}_0} \sum_{x \in \mathcal{X}} C_x M_x^2, \quad g(C) \stackrel{def}{=} \sup_{M \in \mathcal{V}_0} \sum_{x \in \mathcal{X}} C_x M_x.$$

*Assume that $\mathcal{V}_0 = \mathcal{M}_K^{\max} \cap S$, where $S$ is a constraint set that contains $0$ that is closed under the operation of negating any subset of entries. Then $g(\cdot)$ is non-negative and $\forall C \in \mathcal{C}$, there holds*

$$\tilde{g}(C) \le \frac{\pi K}{2} g(C).$$

*Proof.* That $g(\cdot)$ is non-negative follows from the assumption that $0 \in \mathcal{V}_0$. It suffices to show the inequality for $\mathcal{C} = \Delta_{\mathcal{X}}$, since $\forall C \in \left[0, \frac{1}{(m+n)^{3/2}}\right]^{\mathcal{X}}$, we can consider $C' = \frac{C}{\|C\|_1} \in \Delta_{\mathcal{X}}$, and $\tilde{g}(C') \le \frac{\pi K}{2} g(C')$ implies $\tilde{g}(C) \le \frac{\pi K}{2} g(C)$.

By compactness of $\mathcal{V}_0$, $\exists M \in \mathcal{V}_0$ such that the value of $\tilde{g}(C)$ is achieved. Moreover, since $\mathcal{C} = \Delta_{\mathcal{X}}$, $C$ defines a probability distribution on $\mathcal{X}$. Then, it suffices to show that $\exists \tilde{M} \in \mathcal{V}_0$ such that

$$\sum_{x \in \mathcal{X}} C_x M_x^2 = \mathbb{E}_C[M_x^2] \le \frac{\pi K}{2} \mathbb{E}_C[\tilde{M}_x] \le \frac{\pi K}{2} \sum_{x \in \mathcal{X}} C_x \tilde{M}_x.$$

We start the construction of $\tilde{M}$. By assumption that $\mathcal{V} \subseteq \mathcal{M}_K^{\max}$, $\exists U \in \mathbb{R}^{m \times d}, V \in \mathbb{R}^{n \times d}$, $d = \text{rank}(M)$ such that $M = UV^T$ and $\|U\|_{2,\infty}\|V\|_{2,\infty} \le K$. Denote as $u_i, v_j$ the $i$-th and $j$-th row of $U, V$, respectively. Let $S^d$ denote the unit sphere in $\mathbb{R}^d$. Draw a random vector $w \sim S^d$ uniformly at random and consider its inner product with each of the $u_i, v_j$'s. Define $\tilde{U} \in \mathbb{R}^{m \times d}, \tilde{V} \in \mathbb{R}^{n \times d}$ in the following way: with $\tilde{u}_i, \tilde{v}_j$ being the $i$-th and $j$-th row of $\tilde{U}, \tilde{V}$,

$$\tilde{u}_i \stackrel{def}{=} \text{sign}(w^T u_i) u_i, \quad \tilde{v}_j \stackrel{def}{=} \text{sign}(w^T v_j) v_j.$$

Note that $\|\tilde{U}\|_{2,\infty} = \|U\|_{2,\infty}, \|\tilde{V}\|_{2,\infty} = \|V\|_{2,\infty}$. Therefore, together with the assumption that $S$ is closed under negation over any subset of entries, $\tilde{M} \stackrel{def}{=} \tilde{U}\tilde{V}^T \in \mathcal{V}_0$.

Consider the hyperplane parametrized by $w$, $P_w \stackrel{def}{=} \{x \in \mathbb{R}^d \mid w^T x = 0\}$, then

$$\mathbb{P}(\text{sign}(u_i^T v_j) \ne \text{sign}(\tilde{u}_i^T \tilde{v}_j)) = \mathbb{P}(\text{sign}(w^T u_i) \ne \text{sign}(w^T v_j))$$

$$= \mathbb{P}(u_i, v_j \text{ are separated by } P_w) = \frac{\arccos\left(\frac{u_i^T v_j}{\|u_i\|_2 \|v_j\|_2}\right)}{\pi}.$$

Taking expectation over the distribution of the random vector $w$,

$$\mathbb{E}_w[\tilde{u}_i^T \tilde{v}_j] = u_i^T v_j \left(1 - \frac{2\arccos\left(\frac{u_i^T v_j}{\|u_i\|_2 \|v_j\|_2}\right)}{\pi}\right) \ge \frac{2(u_i^T v_j)^2}{\pi \|u_i\|_2 \|v_j\|_2} \ge \frac{2(u_i^T v_j)^2}{\pi k} \iff \mathbb{E}_w[\tilde{M}_{ij}] \ge \frac{2M_{ij}^2}{\pi k}.$$

Taking expectation over distribution $C$ and applying Fubini's Theorem,

$$\mathbb{E}_C[M_x^2] \le \frac{\pi K}{2} \mathbb{E}_w \mathbb{E}_C[\tilde{M}_x],$$

which implies that there exists an instance of $\tilde{M} \in \mathcal{V}_0$ such that

$$\sum_{x \in \mathcal{X}} C_x M_x^2 \le \frac{\pi K}{2} \sum_{x \in \mathcal{X}} C_x \tilde{M}_x.$$

$\square$

**Corollary 25.** *The following inequality holds:*

$$\sup_{M^1,M^2 \in \mathcal{V}} \sum_{x \in \mathcal{X}} C_x(M_x^1 - M_x^2)^2 \leq \pi K \sup_{M^1,M^2 \in \mathcal{V}} \sum_{x \in \mathcal{X}} C_x(M_x^1 - M_x^2),$$

*for max-norm constrained, symmetric version space $\mathcal{V}$ around a given matrix.*

*Proof.* By transformation to a version space $\mathcal{V}_0$ centered around the zero matrix, if $M^1, M^2 \in \mathcal{V}$, then $M \stackrel{\text{def}}{=} \frac{M^1 - M^2}{2} \in \mathcal{V}_0$, where $\mathcal{V}_0 = \mathcal{M}_K^{\max} \cap S$ and $S$ is a constraint set that is closed under negation under any subset of entries. The result is subsumed by Theorem 24. $\square$

## C.2 Proof of regret guarantees

### C.2.1 Proof of Theorem 19

**Theorem 19.** Denote at every time step $t$, consider the concave reward function $r_t(C) \stackrel{\text{def}}{=} H(C) - \alpha G(C, M_t)$. There exists sub-routine gradient-based update $\mathcal{A}_C$ (see Alg. 4 for an example) with the following regret guarantee w.r.t. $r_t$:

1. For setting 1a,

$$\text{Regret}_T^{\mathcal{A}_C} \stackrel{\text{def}}{=} \max_{C \in \mathcal{C}} \sum_{t=1}^{T} r_t(C) - \sum_{t=1}^{T} r_t(C_t) \leq O(\alpha\sqrt{\log(mn)T}) = \tilde{O}(\alpha\sqrt{T}).$$

2. For setting 1b,

$$\text{Regret}_T^{\mathcal{A}_C} \stackrel{\text{def}}{=} \max_{C \in \mathcal{C}} \sum_{t=1}^{T} r_t(C) - \sum_{t=1}^{T} r_t(C_t) \leq O(\alpha\sqrt{(m+n)T}).$$

*Proof of Theorem 19.* We divide the proof into two parts, corresponding to two different choices of $H(\cdot)$ and the corresponding $\mathcal{C}$. Both use online mirror descent (OMD) step as updates. The analysis for the entropy case is slightly more involved due to the gradient behavior at the boundary. We will begin with outlining the algorithm:

**Definition 26** (Bregman divergence). Let $R : \Omega \to \mathbb{R}$ be a continuously-differentiable, strictly convex function defined on a convex set $\Omega$. The *Bregman divergence* associated with $R$ for $p, q \in \Omega$ is defined by

$$B_R(p,q) = R(p) - R(q) - \langle \nabla R(q), p - q \rangle.$$

In particular, Bregman divergence measures the difference between $R(p)$ and the first-order Taylor expansion of $R(p)$ around $q$.

---

**Algorithm 4** $\mathcal{A}_C$

---

1: Input: previous $C_t$, completions $M_t = (M_t^1, M_t^2)$.
2: Require: step-size $\eta$, regularization function $R$.
3: **if** input is empty **then**
4:     Set $(\hat{C}_{t+1})_x = e^{-1}, \forall x \in \mathcal{X}$ in setting 1a, set $\hat{C}_{t+1} = \mathbf{0}_{\mathcal{X}}$ in setting 1b.
5: **else**
6:     Update $\hat{C}_{t+1}$ via $\nabla R(\mathbf{vec}(\hat{C}_{t+1})) = \nabla R(\mathbf{vec}(C_t)) + \eta \nabla r_t(\mathbf{vec}(C_t))$.
7: **end if**
8: Obtain $C_{t+1}$ via Bregman projection: $C_{t+1} = \arg\min_{C \in \mathcal{C}'} B_R(\mathbf{vec}(C), \mathbf{vec}(\hat{C}_{t+1}))$.
9: Output $C_{t+1}$.

---

$\mathcal{C}'$ is taken to be

1. For setting 1a, $\mathcal{C}' = \Delta_{\mathcal{X}}^{\beta} \stackrel{\text{def}}{=} \{C \in \Delta_{\mathcal{X}} : \min_{x \in \mathcal{X}} C_x \geq e^{1-\beta}\}$.

2. For setting 1b, $\mathcal{C}' = \mathcal{C} = \left[0, \frac{1}{(m+n)^{3/2}}\right]^{\mathcal{X}}$, and

The square root diameter $D_R$ of $R(\cdot)$ over $\mathbf{vec}(\mathcal{C}')$ is given by

$$D_R \overset{\text{def}}{=} \sqrt{\max_{X,Y \in \mathcal{C}'}\{R(\mathbf{vec}(X)) - R(\mathbf{vec}(Y))\}}.$$

$\ell_1$**-norm and cube.** In setting 1b, take $R : \mathbf{vec}(\mathcal{C}') \subset \mathbb{R}^{mn} \to \mathbb{R}$ given by $R(x) = \frac{1}{2}\|x\|_2^2$. Then $D_R \le \frac{1}{2\sqrt{m+n}}$. The regret guarantee for online mirror descent also depends on the bound on local norms of the gradients. In particular, the local norm at time $t$ is a function mapping from $\mathbb{R}^{mn}$ to $\mathbb{R}_{++}$ given by

$$\|x\|_t^* = \sqrt{x^\top \nabla^2 R(\tilde{C})x},$$

where $\tilde{C}$ is some convex combination of $C_t$ and $C_{t+1}$ satisfying

$$R(\mathbf{vec}(C_t)) = R(\mathbf{vec}(C_{t+1})) + \nabla R(\mathbf{vec}(C_{t+1}))^\top \mathbf{vec}(C_t - C_{t+1})$$
$$+ \frac{1}{2}\mathbf{vec}(C_t - C_{t+1})^\top \nabla^2 R(\mathbf{vec}(\tilde{C}))\mathbf{vec}(C_t - C_{t+1}).$$

Note that $\forall x \in \mathbf{vec}(\mathcal{C}')$, $\nabla^2 R(x) = I_{mn}$, and thus

$$G_R^2 \overset{\text{def}}{=} \|\mathbf{vec}(\nabla r_t(C_t))\|_t^{*2} = \|\mathbf{vec}(\nabla r_t(C_t))\|_2^2 \le (1 + 2\alpha)^2 mn,$$

where inequality follows from that $(\nabla r_t(C))_x = 1 - \alpha(M_x^1 - M_x^2) \le 1 + 2\alpha$.

By standard Online Mirror Descent (OMD) analysis using the diameter and gradient bounds, by taking $\eta = \frac{D_R}{G_R\sqrt{T}}$, we have the regret bound of $r_t$'s over the cube:

$$\max_{C \in \mathcal{C}'} \sum_{t=1}^{T} r_t(C) - \sum_{t=1}^{T} r_t(C_t) \le D_R G_R \sqrt{T} = O(\alpha\sqrt{(m+n)T}).$$

**Entropy and simplex.** In this setting, take $R(\mathbf{vec}(X)) = -H(X)$, the negative entropy function, $D_R^2 = \log(mn)$. The proof will be divided into two parts: (1) we first show low regret of $\mathcal{A}_C$ w.r.t. $\Delta_{\mathcal{X}}^\beta$, then (2) we show that the best $C$ in $\Delta_{\mathcal{X}}^\beta$ exhibits approximately the same performance as the best $C$ in $\Delta_{\mathcal{X}}$.

Let $\gamma \overset{\text{def}}{=} \beta + 2\alpha$. For any $C \in \Delta_{\mathcal{X}}^\beta$, the gradient of the reward function is bounded by:

$$\|\mathbf{vec}(\nabla r_t(C))\|_\infty \le \|\nabla H(C)\|_\infty + \alpha\|\nabla_C G(C, M_t)\|_\infty \qquad \text{$\Delta$-inequality}$$
$$= \max_{x \in \mathcal{X}} |-1 - \log C_x| + \alpha \cdot \max_{x \in \mathcal{X}} |(M_t^1 - M_t^2)_x| \qquad \text{definition of $\|\cdot\|_\infty, H, G$}$$
$$\le 1 + \log(e^{\beta - 1}) + 2\alpha \qquad C \in \Delta_{\mathcal{X}}^\beta, M_t^1, M_t^2 \in \mathcal{M}_K^{\max}$$
$$\le \gamma.$$

Thus, for some convex combination $\tilde{C}$ of $C_t$ and $C_{t+1}$,

$$G_R^2 = \|\mathbf{vec}(\nabla r_t(C_t))\|_t^{*2} = \sum_{x \in \mathcal{X}} \tilde{C}_x (\nabla r_t(C_t))_x^2 \le \|\mathbf{vec}(\nabla r_t(C_t))\|_\infty^2 \|\tilde{C}\|_1 \le \gamma^2,$$

where the last last inequality follows from $\|\mathbf{vec}(\nabla r_t(C))\|_\infty \le \gamma$ and $\|\tilde{C}\|_1 = 1$.

By standard Online Mirror Descent (OMD) analysis using the diameter and gradient bounds, we have the regret bound of $r_t$'s over the constrained unit simplex $\Delta_{\mathcal{X}}^\beta$ as the following:

$$\max_{C \in \Delta_{\mathcal{X}}^\beta} \sum_{t=1}^{T} r_t(C) - \sum_{t=1}^{T} r_t(C_t) \le D_R G_R \sqrt{T} = O(\alpha\sqrt{\log(mn)T}) = \tilde{O}(\alpha\sqrt{T}),$$

when taking $\eta = \frac{D_R}{G_R\sqrt{T}}$.

With the regret bound established w.r.t. $\Delta_{\mathcal{X}}^\beta$, it is left to show the following inequality, which justifies constraining the feasible set to $\Delta_{\mathcal{X}}^\beta$:

**Lemma 27.** *The following inequality holds for $\beta = 2\log(mn)$ assuming $T = \tilde{O}(m+n)$:*

$$\max_{C \in \Delta_{\mathcal{X}}} \sum_{t=1}^{T} r_t(C) - \max_{C \in \Delta_{\mathcal{X}}^{\beta}} \sum_{t=1}^{T} r_t(C) \leq \tilde{O}\left(\frac{\alpha}{\min\{m,n\}}\right).$$

*Proof of Lemma 27.* Recall $\beta$ fixes an upper-bound on the gradient of the entropy of $C$. This is equivalent to fixing a lower-bound $\delta$ on the entries of $C$, where $\delta \stackrel{\text{def}}{=} e^{1-\beta} = e(mn)^{-2}$. We define $C^*$ as follows:

$$C^* \stackrel{\text{def}}{=} \arg\max_{C \in \Delta_{\mathcal{X}}} \sum_{t=1}^{T} r_t(C).$$

Denote the set of indices where $C^*$ is less than $\delta$ as follows: $S_\delta \stackrel{\text{def}}{=} \{x \in \mathcal{X} \mid C_x^* < \delta\}$. Enumerate $S_\delta$ as $\{(i_k, j_k)\}_{k=1}^{|S_\delta|}$.

Note that $\forall (i_k, j_k) \in S_\delta, \exists (i_k', j_k') \in \mathcal{X}$ such that $C_{i_k', j_k'}^* - \delta \geq \delta - C_{i_k, j_k}^*$. Otherwise, we have by choice of $\delta$,

$$\|C^*\|_1 < mn(2\delta - C_{i_k, j_k}^*) \leq 2\delta mn < 1.$$

We construct the matrix $C$ initialized as $C = C^*$. Next, for each $k \in [1, |S_\delta|]$ we iteratively change two entries in $C$ as follows for all $k$:

1. $C_{i_k, j_k} = \delta$,
2. $C_{i_k', j_k'} = C_{i_k', j_k'}^* - (\delta - C_{i_k, j_k}^*)$.

For each such operation,

$$H(C) - H(C^*) = \left(C_{i_k, j_k} \log\left(\frac{1}{C_{i_k, j_k}}\right) + C_{i_k', j_k'} \log\left(\frac{1}{C_{i_k', j_k'}}\right)\right)$$
$$- \left(C_{i_k, j_k}^* \log\left(\frac{1}{C_{i_k, j_k}^*}\right) + C_{i_k', j_k'}^* \log\left(\frac{1}{C_{i_k', j_k'}^*}\right)\right).$$

Note that $C_{i_k, j_k} + C_{i_k', j_k'} = C_{i_k, j_k}^* + C_{i_k', j_k'}^* \geq 2\delta$, and $C_{i_k, j_k}^* < \delta$. Thus, $H(C) - H(C^*) \geq 0$ holds for each operation. $C$ constructed by this enumeration satisfies $H(C) \geq H(C^*)$. On the other hand, $G$ is Lipschitz. In particular, let $\bar{M}^1 = \frac{1}{T}\sum_{t=1}^{T} M_t^1$, $\bar{M}^2 = \frac{1}{T}\sum_{t=1}^{T} M_t^2$, then

$$\sum_{t=1}^{T} G(C, M_t^1, M_t^2) - \sum_{t=1}^{T} G(C^*, M_t^1, M_t^2) = \alpha T \sum_{i,j} (C_{ij} - C_{ij}^*)(\bar{M}_{ij}^1 - \bar{M}_{ij}^2) \qquad \text{linearity of } G$$

$$\leq 2\alpha T \|C - C^*\|_1 \qquad \bar{M}^1, \bar{M}^2 \in [-1,1]^{\mathcal{X}}$$
$$\leq 4mn\delta\alpha T \qquad \|C - C^*\|_\infty \leq \delta$$
$$= \frac{4e\alpha T}{mn}. \qquad \delta = \frac{e}{(mn)^2}$$

Under the assumption that $T \geq \tilde{O}(m+n)$, we conclude that

$$\max_{C \in \Delta_{\mathcal{X}}} \sum_{t=1}^{T} r_t(C) - \max_{C \in \Delta_{\mathcal{X}}^{\beta}} \sum_{t=1}^{T} r_t(C) \leq \tilde{O}\left(\frac{\alpha}{\min\{m,n\}}\right).$$

$\square$

By taking $\beta = O(\log(mn))$ and assuming $T = \tilde{O}(m+n)$, we can conclude that

$$\text{Regret}_T^{\mathcal{A}_C} \stackrel{\text{def}}{=} \max_{C \in \Delta_{\mathcal{X}}} \sum_{t=1}^{T} r_t(C) - \sum_{t=1}^{T} r_t(C_t) \leq O(\alpha\sqrt{\log(mn)T}) = \tilde{O}(\alpha\sqrt{T}).$$

$\square$

**Theorem 20.** Denote at every time step $t$, consider the concave reward function $\gamma_t(M) \overset{\text{def}}{=} G(C_t, M) - \theta f_t(M)$. There exists sub-routine gradient-based update $\mathcal{A}_M$ (see Alg. 5 for an example) such that, under either setting 1a or 1b, the following regret guarantee w.r.t. $\gamma_t$ holds:

$$\text{Regret}_T^{\mathcal{A}_M} \overset{\text{def}}{=} \max_{M \in \mathcal{M}_K^{\max} \times \mathcal{M}_K^{\max}} \sum_{t=1}^{T} \gamma_t(M) - \sum_{t=1}^{T} \gamma_t(M_t) \leq O(K\theta\sqrt{(m+n)T}).$$

*Proof of Theorem 20.* We will begin by outlining the update algorithm $\mathcal{A}_M$, then introduce the key definitions used in $\mathcal{A}_M$, and proceed to prove Theorem 20. We note that this algorithm is modified from the matrix multiplicative weights for online matrix prediction (Algorithm 2) in Hazan et al. [2012b].

---

**Algorithm 5** $\mathcal{A}_M$

---

1: Input: $C_t, M_t = (M_t^1, M_t^2) \in \mathcal{M}_K^{\max}$, $(i_t, j_t, o_t)$.
2: **if** input is empty **then**
3:    Output: $\phi^{-1}\left(\frac{K}{2}I\right)$.
4: **end if**
5: Compute $X_t = \phi(M_t^1, M_t^2)$ with $\phi(\emptyset) = \frac{K}{2}I$.
6: Create matrix $L_t(\gamma_t)$ according to Definition 29.
7: Update: with step size $\eta = \frac{1}{2(1+8\theta)}\sqrt{\frac{(m+n)\log(2p)}{T}}$, project w.r.t. matrix relative entropy:

$$X_{t+1} = \arg\min_{X \in \mathcal{K}_X} \Delta(X, \exp(\log(X_t) + \eta L_t(\gamma_t))).$$

8: Output: $M_{t+1} = (M_{t+1}^1, M_{t+1}^2) = \phi^{-1}(X_{t+1})$.

---

$\mathcal{K}_X$ is given by

$$\mathcal{K}_X \overset{\text{def}}{=} \{X \in Sym(2p) : X \succeq 0, \ \text{Tr}(X) \leq 2K(m+n), \ X_{ii} \leq K, \ X[:p,:p] - X[p:,p:] \in [-1,1]^{p \times p}\}.$$

The operator $\phi$ is given as the following:

**Lemma 28.** $\forall M^1, M^2 \in \mathcal{M}_K^{\max}$, $M = \begin{bmatrix} M^1 & \mathbf{0} \\ \mathbf{0} & M^2 \end{bmatrix}$ *is* $(K, 2K(m+n))$*-decomposable.*

Denote $p \overset{\text{def}}{=} 2(m+n)$. $(\beta, \tau)$-decomposability allows for two matrices in $\mathcal{M}_K^{\max}$ to be embedded in $Sym(2p)$:

$\phi(\cdot, \cdot) : \mathcal{M}_K^{\max} \times \mathcal{M}_K^{\max} \to Sym(2p)$ is the embedding operator given by

$$\phi(M) \overset{\text{def}}{=} \phi(M^1, M^2) \overset{\text{def}}{=} \begin{bmatrix} P & \mathbf{0} \\ \mathbf{0} & N \end{bmatrix},$$

where $P, N$ are the PSD matrices given by the $(\beta, \tau)$-decomposition.

The descent matrix $L_t(\gamma_t)$, which we shorthand denote as $L_t$, is constructed as the following:

**Definition 29** (Descent matrix). At time $t$, we define the matrix $L_t \in Sym(2p)$ as $L_t \overset{\text{def}}{=} L_t^G + L_t^F$, where $L_t^G$ is symmetric and

$$\begin{cases} L_t^G[1:m, 2m+1:2m+n] = C_t \\ L_t^G[p+m+1:p+2m, p+2m+n+1:2p] = C_t \\ L_t^G[m+1:2m, 2m+n+1:p] = -C_t \\ L_t^G[p+1:p+m, p+2m+1:p+2m+n] = -C_t \\ L_t^G[i,j] = 0 \text{ if otherwise and } j \geq i \end{cases}$$

$L_t^F$ is symmetric and

$$\begin{cases} L_t^F[i_t, 2m+j_t] = -2\theta((M_t^1)_{i_t,j_t} - o_t), L_t^F[m+i_t, 2m+n+j_t] = -2\theta((M_t^2)_{i_t,j_t} - o_t) \\ L_t^F[p+i_t, p+2m+j_t] = 2\theta((M_t^1)_{i_t,j_t} - o_t), L_t^F[p+m+i_t, p+2m+n+j_t] = 2\theta((M_t^2)_{i_t,j_t} - o_t) \\ L_t^F[i,j] = 0 \text{ if otherwise and } j \geq i \end{cases}$$

Note that by construction $(L_t^G)^2$, $(L_t^F)^2$ are diagonal matrices, and $\mathrm{Tr}(L_t^2) \leq O(\theta^2)$.

The rest of the proof follows from Section 3.2 in [Hazan et al., 2012b].

$\square$

## C.3 Proof of offline implications

### C.3.1 Proof of Lemma 21

**Lemma 21.** After $T$ iterations, and assume that for some $\delta > 0$,

$$\frac{1}{\theta}\left(2D + \frac{\mathrm{Regret}_T^{\mathcal{A}_M}}{T}\right) \leq \frac{\delta^{2/3}}{2},$$

with $D = 1$ in setting 1a, and $D = \sqrt{m+n}$ in setting 1b. The following properties hold on the obtained $\bar{C} \overset{\text{def}}{=} \frac{1}{T}\sum_{t=1}^T C_t$ returned by Algorithm 3: with probability $\geq 1 - \exp\left(-\frac{\delta^{4/3}T}{512}\right)$,

$$H(\bar{C}) - \max_{M \in \mathcal{V}_{T,\delta}^2} \alpha \cdot G(\bar{C}, M) \geq \max_{C \in \mathcal{C}} \min_{M \in \mathcal{V}_{\delta^{2/3}}^2} \{H(C) - \alpha \cdot G(C, M)\} - 2\alpha\theta\delta - \frac{B_T}{T}.$$

*Proof of Lemma 21.* For notation convenience, denote

$$C^\star, M_\star^1, M_\star^2 = \arg\max_{C \in \mathcal{C}} \arg\min_{(M^1, M^2) \in \mathcal{V}_{\delta^{2/3}}^2} H(C) - \alpha G(C, M^1, M^2).$$

Consider the subroutine $\mathcal{A}_M$. Under the assumption, $\mathcal{A}_M$ is a low-regret OCO algorithm for $\gamma_t$'s. In particular, under the realizable assumption, since there exist $M = (M^1, M^2)$ such that $f_t(M) = \gamma_t(M) = 0, \forall t$, we have that for the sequence of $M_t$'s output by the algorithm,

$$\frac{1}{T}\sum_{t=1}^T f_t(M_t) = \frac{1}{T\theta}\sum_{t=1}^T (G(C_t, M_t) - \gamma_t(M_t)) \leq \frac{1}{\theta}\left(2D + \frac{\mathrm{Regret}_T^{\mathcal{A}_M}}{T}\right) \leq \frac{\delta^{2/3}}{2}.$$

Note that $\forall M \in \mathcal{M}_K^{\max}$, $M$ can be seen as a function mapping from $\mathcal{X}$ to $[-1, 1]$. Denote $f(M) \overset{\text{def}}{=} \mathbb{E}_{(x,o)\sim\mathcal{D}}[(M_x^1 - o)^2 + (M_x^2 - o)^2]$. Define $Z_t = f(M_t) - f_t(M_t)$, $X_t = \sum_{i=1}^t Z_i$, then we have with $\mathcal{F}_t$ denoting the filtration generated by the algorithm's randomness up to iteration $t$, and since $M_t \in \mathcal{F}_{t-1}$,

$$\mathbb{E}[Z_t \mid \mathcal{F}_{t-1}] = 0, \ \mathbb{E}[X_t \mid \mathcal{F}_{t-1}] = X_{t-1},$$

and $|X_t - X_{t-1}| = |Z_t| \leq 8$. By Azuma's inequality, we have $\forall \varepsilon > 0$,

$$\mathbb{P}\left(\frac{1}{T}\sum_{t=1}^T f(M_t) - f_t(M_t) > \sqrt{\frac{128\log\left(\frac{1}{\varepsilon}\right)}{T}}\right) = \mathbb{P}\left(\frac{1}{T}\sum_{t=1}^T X_t > \sqrt{\frac{128\log\left(\frac{1}{\varepsilon}\right)}{T}}\right) \leq \varepsilon.$$

We can conclude that with probability at least $1 - \exp\left(-\frac{\delta^{4/3}T}{512}\right)$,

$$f(\bar{M}) \leq \frac{1}{T}\sum_{t=1}^T f(M_t) \leq \frac{1}{T}\sum_{t=1}^T f_t(M_t) + \frac{\delta^{2/3}}{2} \leq \frac{\delta^{2/3}}{2} + \frac{\delta^{2/3}}{2} = \delta^{\frac{2}{3}},$$

in which case $\bar{M}^1, \bar{M}^2 \in \mathcal{V}_{\delta^{2/3}}$. We have thus with probability at least $1 - \exp\left(-\frac{\delta^{4/3}T}{512}\right)$,

$$
\begin{aligned}
&H(C^\star) - \alpha G(C^\star, M_\star^1, M_\star^2) \\
&\leq H(C^\star) - \alpha G(C^\star, \bar{M}^1, \bar{M}^2) && (M_\star^1, M_\star^2) \text{ are optimal w.r.t. } C^\star \text{ in } \mathcal{V}_\delta \\
&= \frac{1}{T}\sum_{t=1}^{T} r_t(C^\star) && \text{linearity of } G, \text{ definition of } r_t \\
&\leq \frac{1}{T}\sum_{t=1}^{T} r_t(C_t) + \frac{\text{Regret}_T^{\mathcal{A}_C}}{T} && \mathcal{A}_C \text{ regret guarantee} \\
&\leq H(\bar{C}) - \frac{\alpha}{T}\sum_{t=1}^{T} G(C_t, M_t^1, M_t^2) + \frac{\text{Regret}_T^{\mathcal{A}_C}}{T} && \text{concavity of } H(\cdot) \text{ on } \mathcal{C} \\
&\leq H(\bar{C}) - \frac{\alpha}{T}\sum_{t=1}^{T} \gamma_t(M_t^1, M_t^2) + \frac{\text{Regret}_T^{\mathcal{A}_C}}{T} && f_t \geq 0 \\
&\leq H(\bar{C}) - \frac{\alpha}{T}\sum_{t=1}^{T} \gamma_t(\hat{M}^1, \hat{M}^2) + \frac{B_T}{T} && \forall (\hat{M}^1, \hat{M}^2) \in \mathcal{M}_K^{\text{max}2} \text{ by } \mathcal{A}_M \text{ regret guarantee} \\
&\leq H(\bar{C}) - \max_{(M^1, M^2) \in \mathcal{V}_{T,\delta}^2} \alpha \cdot G(\bar{C}, M^1, M^2) + 2\alpha\theta\delta + \frac{B_T}{T} && \text{definition of } \mathcal{V}_{T,\delta}
\end{aligned}
$$

$\square$

### C.3.2 Proof of Theorem 22

**Theorem 22.** Suppose the underlying sampling distribution is $\mu$. Let $C_\mu \in \mathcal{C}$ be its corresponding confidence matrix. In particular, for setting 1a, $C_\mu = \mu$, i.e. $(C_\mu)_{ij} = \mathbb{P}_\mu((i,j) \text{ is sampled})$; for setting 1b, $C_\mu$ satisfies that $C_\mu/\|C_\mu\|_1 = \mu$. Then, for any $\delta > 0$, Algorithm 3 run with $\alpha = \delta^{-1/6}$ returns a $\bar{C}$ that guarantees the following bounds: with probability $\geq 1 - \exp\left(-\frac{\delta^2 T}{128}\right) - \exp\left(-\frac{\delta^{4/3}T}{512}\right)$,

1. For setting 1a, take $\theta = 4\delta^{-2/3}$, after $T = \tilde{O}(\delta^{-2}K^2(m+n))$ iterations,
   (a) $H(\bar{C}) \geq H(C_\mu) - O(\delta^{1/6})$.
   (b) $\max_{M^1, M^2 \in \mathcal{V}_{\frac{\delta}{2}}} G(\bar{C}, M^1, M^2) \leq O(\delta^{1/6} \log(mn))$.

2. For setting 1b, take $\theta = 4\delta^{-2/3}\sqrt{m+n}$, after $T = \tilde{O}(\delta^{-2}K^2(m+n))$ iterations,
   (a) $\|\bar{C}\|_1 \geq \|C_\mu\|_1 - O(\delta^{1/6}\sqrt{m+n})$.
   (b) $\max_{M^1, M^2 \in \mathcal{V}_{\frac{\delta}{2}}} G(\bar{C}, M^1, M^2) \leq O(\delta^{1/6}\sqrt{m+n})$.

*Proof of Theorem 22.* First, note that $\forall i, j, o$, we have by assumption $(M_{ij} - o)^2 \in [0, 4]$, $\forall M \in \mathcal{M}_K^{\text{max}}$. Therefore, by subgaussian concentration, $\forall c \geq 1$,

$$
\mathbb{P}\left(\frac{1}{T}\sum_{t=1}^{T}(M_{i_t,j_t} - o_t)^2 - \mathbb{E}_{i,j}[(M_{ij} - o)^2] \geq \frac{\delta}{2}\right) \leq \exp\left(-\frac{\delta^2 T}{128}\right).
$$

Therefore, with probability at least $1 - \exp\left(-\frac{\delta^2 T}{128}\right)$, we have

$$
\mathcal{V}_{\frac{\delta}{2}} \subseteq \mathcal{V}_{T,\delta}.
$$

Therefore, it suffices to show the inequality in (b) for the maximum over $\mathcal{V}_{T,\delta}$. Let

$$C^\star, M^1_\star, M^2_\star = \arg\max_{C \in \mathcal{C}} \quad \arg\min_{(M^1, M^2) \in \mathcal{V}^2_{\delta^{2/3}}} \quad H(C) - \alpha G(C, M^1, M^2)$$

$$= \arg\max_{C \in \mathcal{C}} \quad \arg\min_{(M^1, M^2) \in \mathcal{V}^2_{\delta^{2/3}}} \quad H(C) - \delta^{-1/6} G(C, M^1, M^2).$$

Choose $T$ such that $\frac{B_T}{T} \leq \alpha\theta\delta$. Note that in both settings, the choice of $\theta$ and $T$ satisfies the assumption in Theorem 21.

**Simplex and entropy.** We can bound $G(C_\mu, M^1_\star, M^2_\star)$ by

$$G(C_\mu, M^1_\star, M^2_\star) = \mathbb{E}_{x \sim \mu}[(M^1_\star)_x - (M^2_\star)_x]$$

$$= \sqrt{(\mathbb{E}_{(x,o) \sim \mathcal{D}}[(M^1_\star)_x - o_x] - \mathbb{E}_{(x,o) \sim \mathcal{D}}[(M^2_\star)_x - o_x])^2}$$

$$\leq \sqrt{2(\mathbb{E}_{(x,o) \sim \mathcal{D}}[(M^1_\star)_x - o_x]^2 + \mathbb{E}_{(x,o) \sim \mathcal{D}}[(M^2_\star)_x - o_x]^2)} \qquad (a-b)^2 \leq 2(a^2+b^2)$$

$$\leq \sqrt{2(\mathbb{E}_{(x,o) \sim \mathcal{D}}[((M^1_\star)_x - o_x)^2] + \mathbb{E}_{(x,o) \sim \mathcal{D}}[((M^2_\star)_x - o_x)^2])} \qquad \text{Jensen's}$$

$$\leq 2\delta^{1/3} \qquad\qquad M^1_\star, M^2_\star \in \mathcal{V}_{\delta^{2/3}}$$

Theorem 21 implies that with probability $\geq 1 - \exp\left(-\frac{\delta^{4/3}T}{512}\right)$,

$$H(\bar{C}) - \max_{M^1, M^2 \in \mathcal{V}_{T,\delta}} \delta^{-1/6} G(\bar{C}, M^1, M^2) \geq H(C_\mu) - \alpha G(C_\mu, M^1_\star, M^2_\star) - 2\alpha\theta\delta - \frac{B_T}{T}$$

$$\geq H(C_\mu) - 12\delta^{1/6},$$

Note that by definition $\max_{M^1, M^2 \in \mathcal{V}_{T,\delta}} G(\bar{C}, M^1, M^2) \geq 0$, and thus

$$H(\bar{C}) \geq H(C_\mu) - 12\delta^{1/6}.$$

Since $H(\cdot)$ is bounded by $[0, \log(mn)]$ over $\mathcal{C}$, then

$$\max_{M^1, M^2 \in \mathcal{V}_{T,\delta}} G(\bar{C}, M^1, M^2) \leq \delta^{1/6}\left(H(\bar{C}) - H(C_\mu)\right) + 12\delta^{1/3}$$

$$\leq \delta^{1/6}\log(mn) + 12\delta^{1/3}$$

$$\leq O(\delta^{1/6}\log(mn)).$$

**Cube and $\ell_1$ norm.** We can bound $G(C_\mu, M^1_\star, M^2_\star)$ by

$$G(C_\mu, M^1_\star, M^2_\star) \leq \sqrt{m+n}\,\mathbb{E}_{x \sim \mu}\left[(M^1_\star)_x - (M^2_\star)_x\right] \leq 2\delta^{1/3}\sqrt{m+n}.$$

Theorem 21 implies that

$$H(\bar{C}) - \max_{M^1, M^2 \in \mathcal{V}_{T,\delta}} \delta^{-1/6} G(\bar{C}, M^1, M^2) \geq H(C_\mu) - 12\delta^{1/6}\sqrt{m+n}.$$

Thus, similar as before, we get

$$H(\bar{C}) \geq H(C_\mu) - 12\delta^{1/6}\sqrt{m+n},$$

and

$$\max_{M^1, M^2 \in \mathcal{V}_{T,\delta}} G(\bar{C}, M^1, M^2) \leq \delta^{1/6}\left(H(\bar{C}) - H(C_\mu)\right) + 12\delta^{1/3}\sqrt{m+n}$$

$$\leq O(\delta^{1/6}\sqrt{m+n}).$$

$\square$

# D Implementation Details

With the theoretical guarantee established above, we implemented a simple version of ODD in Section 6 and experimented with both simulated toy datasets and real-world datasets including the well-known MovieLens dataset. In particular, the simplification of ODD we implemented is given by the following algorithm:

---

**Algorithm 6** Simplified ODD

---

1: **Input**: initial (uniform) distribution $C_1$, matrix $M \in \mathcal{K}$, parameters $\eta > 0$, $\alpha > 0$, matrix prediction update function `matrix-predict`$(\cdot)$. $R(\cdot) = -H(\cdot)$.
2: **for** $t = 1, 2, ..., T$ **do**
3:     Adversary draws tuple $x_t = (i_t, j_t)$, reveals $M_{x_t}$.
4:     Update $M_{t+1} = $ `matrix-predict`$(C_t, M_t)$.
5:     Consider reward $\tilde{r}_t(C) \overset{\text{def}}{=} \alpha H(C) - \langle C, M_t \rangle$.
6:     Update $\nabla R(\hat{C}_{t+1}) \leftarrow \nabla R(C_t) + \eta \nabla \tilde{r}_t(C_t)$.
7:     Project $C_{t+1} = \underset{C \in \Delta_\mathcal{X}}{\arg\min} \|C - \hat{C}_{t+1}\|_{\text{fr}}$.
8: **end for**
9: **return**: $\bar{C} \overset{\text{def}}{=} \frac{1}{T} \sum_{t=1}^{T} C_t$.

---

This algorithm is an instantiation of our online algorithm ODD (Algorithm 3), using mirror descent for updating the confidence matrix, and an arbitrary matrix completion method called `matrix-predict`.

# E   Definitions

**Matrix logarithm.**   Given the matrix $X \in Sym(n)$, $X \succeq 0$, $X$ admits a diagonalization $X = V\Sigma V^T$, where $\Sigma$ can be written as follows:

$$\Sigma \stackrel{\text{def}}{=} \begin{bmatrix} \Sigma_{11} & 0 & ... & 0 \\ 0 & \Sigma_{22} & ... & 0 \\ ... & ... & ... & ... \\ 0 & 0 & ... & \Sigma_{nn} \end{bmatrix}.$$

The logarithm of $X$ is given by:

$$\log(X) \stackrel{\text{def}}{=} V \begin{bmatrix} \log(\Sigma_{11}) & 0 & ... & 0 \\ 0 & \log(\Sigma_{22}) & ... & 0 \\ ... & ... & ... & ... \\ 0 & 0 & ... & \log(\Sigma_{nn}) \end{bmatrix} V^T.$$

Matrix logarithm satisfies the following properties for $X, Y \in Sym(n)$, $X, Y \succ 0$:

1. $\text{Tr}(\log(XY)) = \text{Tr}(\log(X)) + \text{Tr}(\log(Y))$.
2. If $XY = YX$, then $\log(XY) = \log(X) + \log(Y)$.
3. $\log(cI) = (\log c)I, \forall c > 0$.

**Matrix relative entropy.**   Given matrices $X, Y \in Sym(n)$, $X, Y \succeq 0$, their relative entropy is given by

$$\Delta(X, Y) \stackrel{\text{def}}{=} \text{Tr}(X \log(X) - X \log(Y) - X + Y).$$

**$(\beta, \tau)$-decomposability.**   $M \in \mathbb{R}^{m \times n}$ is $(\beta, \tau)$-decomposable if $\exists P, N \in Sym(m+n)$, $P, N \succeq 0$:

$$P - N = \begin{bmatrix} \mathbf{0} & M \\ M^T & \mathbf{0} \end{bmatrix}, \quad \text{Tr}(P) + \text{Tr}(N) \leq \tau, \quad \max_i P_{ii}, \max_i N_{ii} \leq \beta.$$