# OpenReview forum: "Partial Matrix Completion"
_NeurIPS.cc/2023/Conference — NeurIPS 2023 poster_

### Official Review · Reviewer_wLHP · 2023-06-29

**Soundness:** 3 good
**Presentation:** 3 good
**Contribution:** 4 excellent
**Rating:** 7
**Confidence:** 4

**Summary:**

This paper introduces the learning problem of simultaneously estimating an unknown ground truth matrix based on the observed entries and providing a set of weights/confidence scores for each unknown entry with two properties: (1) the RMSE, weighted by the confidence scores, satisfies generalization bounds analogous to the standard results in the classic setting, (2) the confidence scores have sufficient *coverage*, i.e. sufficiently many entries have sufficiently high confidence scores. Thus, in light of the existing distribution-free results, this can be viewed as a targeted joint estimation of the ground truth matrix and the ground truth sampling distribution over entries. The main theorems in this first direction are Theorems 8 and 10:

Theorem 8 shows that Algorithm 1, which relies on the intractable optimization problem "MP 3", achieves a sample complexity similar to existing state-of-the-art distribution-free results for both the max norm and the trace norm constraints, with the coverage of the matrix of confidence scores $C$ being at least as large as that of the ground truth sampling distribution. The key aspect of the algorithm is to solve the dual problem involved in MP3, which utilizes the following powerful observation (cf. lines 534 and 227-230): if a matrix of confidence scores $C$ guarantees that any matrix with small empirical $L^2$ norm (over the i.i.d. training sample) must also have small weighted $L^2$ norm w.r.t. $C$ will also guarantee that any matrix with small empirical $L^2$-deviation from a ground truth matrix will also have small $C$-weighted $L^2$ distance to the ground truth matrix. This simple observation is enough to ensure that the construction of the confidence scores $C$ only depends on the sampled entries, and not on the observed ratings. The algorithm from Theorem 8 is shown to be NP-hard.

Theorem 10 shows a slightly worse guarantee  ($1/\epsilon^4$ instead of $1/\epsilon^2$ due to going back and forth between $L^2$ and $L^1$ losses via Jensen's inequality as well as novel results) for an improved algorithm which replaces the $L^2$ loss in MP3 by a linear loss function. The proof of the result relies on a remarkable and original result which is proved in the main text via the probabilistic method (c.f. Proposition 11). The algorithm is tractable, but the result only applies to the max norm, not the trace norm.

In the next part of the paper (i.e., Appendix B which is naturally part of the main paper and whose own appendix is Appendix C), the authors study the online learning setting, providing sublinear regret bounds for both algorithms in Theorems 18 and 19 (summarized in Corollary 17). Those results are valid under an adversarial sampling regime and apply to regret as defined by the loss function $h_t$ defined in line 656. The results are more favorable in terms of dependence on the size of the matrix than the results of [1], but not immediately comparable due to subtle differences in the definition of regret. Note that function $h_t$ is not uniquely defined. Rather, it has two possible definitions based on the choice of function $H(C)$, which lead to a different treatment (c.f. point 1.a and 1.b in lines 658 to 660). The proof techniques are a mix of techniques modified from [1] and [2] and more completely original methods (e.g. the proof of Theorem 23 (which is an analog of Proposition 11 in the online context, and the proof of Lemma 25).

Finally, the authors empirically evaluate their method on a small-scale semi-synthetic dataset consisting of $250$ most popular users and items in MovieLens by comparing the 'confidence level' with the test error. Although it is not explicitly stated, the confidence level most likely refers to $\sum_{x\in\mathcal{X}} C_x$ (the sum of all the confidence scores at all entries).






========Post-rebuttal========

The authors have satisfactorily addressed most of my concerns, and I am still convinced that the paper is very high quality and I will keep my score. For the benefit of the community, I hope the authors polish the issues raised.


=========



**References**

[1] Elad Hazan, Satyen Kale, and Shai Shalev-Shwartz, "Near-optimal algorithms for online matrix prediction", COLT 2012.

[2] Elad Hazan et al. "Introduction to online convex optimization." Foundations and trends in optimization, 2016.

[3] Nathan Srebro, "Rank, Trace-Norm and Max-Norm", COLT 2005.

[4] Prateek Jain, Soumyabrata Pal Online Low Rank Matrix Completion. ICLR 2023.


[5]  Yuxin Chen, Yuejie Chi, Jianqing Fan, Cong Ma, Yuling Yan. Noisy Matrix Completion: Understanding Statistical Guarantees for Convex Relaxation via Nonconvex Optimization.  SIAM journal of Optimization, 2020.

**Strengths:**

This is an outstanding contribution:

1. To the best of my knowledge, the results presented in this paper are **highly original and impactful**. Many of the proofs require significant innovation and are non-trivial. The setting presented in the first part of the paper is an exciting new paradigm and the observations implicit in the results are very deep. I find it especially fascinating how the function class restriction on the set of predicted matrices is enough to allow us to construct the matrix $C$ (which is, after all, an estimate of the sampling distribution) without making any explicit assumptions or restrictions on the class to which the sampling distribution belongs. This is presumably because there is some implicit equivalence relation between distributions in terms of how their corresponding generalization gaps are evaluated on ground truth matrices inside a restricted class.

2. The online learning results are of great theoretical and practical importance and are **highly nontrivial** to prove.

**Weaknesses:**

Note: Overall, this is still an **extremely interesting and impactful paper** despite the limitations in terms of clarity described below.


**Main weaknesses/Summary:**
 My main problem with the paper is that it is not very self-contained and could be presented better/more polished:

 **1** There are several minor errors in the supplementary. In particular, to the best of my knowledge, the **proof of Lemma 3** in Appendix A1 is **wrong**: the equation on line 476.5 is incorrect as it ignores cross-terms. The result is still correct, for different reasons: in the case of the nuclear norm, this is a classic result that can be shown without recourse to the factorized decomposition of the matrix. In the case of the max norm, it follows from the expression of the ball w.r.t. the max norm as the convex envelope of the set of rank 1 matrices (c.f. [3]).
(*Please fix this*)

 **2**.  The explanations are not always perfectly self-contained and assumptions are not always well introduced.

**2.a**  For instance, Theorem 18 requires a condition on the learning rate $\eta$ which is not stated in the theorem. It is clearly needed from applying what the authors call "standard Online Mirror Descent analysis", and it appears in line 763.  It wouldn't hurt to add a complete citation and a restatement of the known result with more detailed calculations. (I found one at https://www.cs.cornell.edu/courses/cs6783/2019fa/lec17.pdf for instance). Note also that it seems that $\|.\|_t^{*}$ is undefined and $\tilde{C}$ doesn't appear in the formula below line 757 which makes it strange that it is introduced when presenting this equation. The same problem is on line

**2.b** It is also not immediately apparent whether Corollary 21 (appendix, page 21) requires the sampling distribution to be uniform (as stated in the preamble to the Corollary (lines 702 to 703) or if it is an arbitrary distribution $\mu$ as stated in the first line of the corollary itself.


**2.c** I found the proof of the NP-hardness somewhat confusing. See questions.

**3** It seems like the matrix relative entropy from line 860 is not actually used in the paper (and certainly not in the definition of the regret/loss function $h_t$ in line 656 (even in the entropy case 1.a, this is an elementwise entropy.  I am not sure but I think it is used in [1]. A more detailed comparison of the definitions of regret in both places would be nice.




**Minor issues/mathematical typos:**

1. In Theorem 8, $\mu_{\max}$ is used in line 517 but is only defined later in Proposition 16 on page 16.

2. In line 65 in the main paper, I it should be $\tilde{O}(|U|)$ rather than $\simeq |U|$ since there is definitely a log term involved.

3.(minor) Algorithm 3 is only understandable when considered in combination with the algorithms that define $\mathcal{A}_C$ (line 751 page 24) and $\mathcal{A}_M$ (line 799 page 27). Algorithm 4 (definition of $\mathcal{A}_C$ itself depends on the quantity $R$ which is only defined later in line 756. In particular, algorithm 3 requires some hyperparameters such as $\eta$ from Algorithm 5 and $\theta$ from algorithm 3, making it difficult to read in the order in which it is presented.

4. It seems like the initialization of $\hat{C}$ in line 4 of Algorithm 4 is incorrect (missing normalization step to make the matrix belong to the simplex).

5. The notion of Bregman projection and the associated notation $B_R$ is not defined. It would help the reader a lot to finish line 8 with an additional  $=argmin_{c\in\mathcal{C}'} R(C) -R( {\hat{C}}_{t+1})$ $-\langle$  $\nabla$ $ R(\hat{C}_{t+1}),C-\hat{C}_{t+1}\rangle $ [ apologies about the markdown not working properly], which probably can be done without adding a line.

6. It seems like the settings 1.a and 1.b are described in the wrong order (swapped) on lines 753 and 754.

7. In the caption of Figure 1, stochastic block models are mentioned in a somewhat tangential way.

8.It seems like the "matrix completion and recommendation system"  part of the related works is missing relevant works. Probably most notably [5] and [4].

9. I think there is a factor of 2 missing in equation (6) (line 300.5). This error is not present  (the factor of 2 is there) in the proof of Theorem 23 in line 732.

10. Iines 684 and 744, there are missing references ("see section ?? for an example")

11. In line 149, I think the expectation should not run over $x\sim\mathcal{X}$ but instead over $x\sim \nu$.


12. There is an equal sign missing in the equation on line 867.5

13. There is a missing $\leq \epsilon$ at the end of line 207



14. Line 179, I think the square loss is actually 4-Lipschitz, not 2-Lipschitz, when both of its arguments are in $[-1,1]$


15. I think there is a factor of $\alpha$ missing in the right hand side of the inequality in Lemma 25.




**Typos and extremely minor issues:**


1. I found the use of "$\Delta$-inequality to mean triangle inequality in line 764 quite confusing given that $\nabla$ is a loaded notation: cf. the set $\nabla_{\chi}^\beta$ and perhaps more confusingly the matrix relative entropy from line 860 on page 32.

2. Line 755 is not a complete sentence.

4. It would be nice to define "negations" in lemma 3 (in addition to fixing the proof).

5. There shouldn't be a capital letter at "Let" at the top of page 15.

6. it would be nice to mention again the stability w.r.t. negations in line 534 as the argument is somewhat key.

7. Missing periods in equations (8) and (9)  and in line 782.

8. Extra "and" in line 723.

9. An extra "That" at the beginning of the first sentence of the proof of Theorem 23 on page 23 is required to make a sentence.

10. Missing determinant in line 744.

11. Line 764 "approximately same performance", missing "the".

12. Line 788 "assume" should be "assuming"

13. 812: "notation convenience" should be "notational convenience"














**References**

[1] Elad Hazan, Satyen Kale, and Shai Shalev-Shwartz, "Near-optimal algorithms for online matrix prediction", COLT 2012.

[2] Elad Hazan et al. "Introduction to online convex optimization." Foundations and trends in optimization, 2016.

[3] Nathan Srebro, "Rank, Trace-Norm and Max-Norm", COLT 2005.

[4] Prateek Jain, Soumyabrata Pal Online Low Rank Matrix Completion. ICLR 2023.

[5]  Yuxin Chen, Yuejie Chi, Jianqing Fan, Cong Ma, Yuling Yan. Noisy Matrix Completion: Understanding Statistical Guarantees for Convex Relaxation via Nonconvex Optimization.  SIAM journal of Optimization, 2020.

**Questions:**

In decreasing order of importance:

**1.** The proof of NP-hardness contains some confusing elements for me. I apologize in advance if this is due to my lack of familiarity with the related results:

**1.a** I  don't understand why problem (7) is equivalent to problem MP 3. Indeed, I can't see why the PSD condition appears. I have a feeling this comes from the ($\beta$,$\tau$) decomposability, but there is definitely an argument missing.

**1.b** The first argument in the "soundness" part which states that we can assume $X$ is rank one w.l.o.g. I also don't understand this argument: why is it the case that the matrix $\tilde{X}$ achieves value $k^2$ if the matrix $X$ does. This is absolutely not obvious to me.
I agree with the rest of the proof assuming this is correct.


**2** Can you clarify whether the expectation in the equation on line 603.5 (bottom of page 18) refers only to the randomness in the choice of the loss function $h_t$ (which is assumed to be "stochastically i.i.d.). In that case, shouldn't there also be an expectation on the right hand side in front of the regrets?

**2.2** The lengthy discussion at the beginning of Section B culminating in the equation from line 603.5 seems to mostly justify the validity of the Game-Regret (see line 679) as a measure of performance of the algorithm by linking it to the minimax regret. Why not apply the result from line 603.5 and make a self-contained theorem which applies to the minimax regret in the paper's setting?


**3** On lines 786-788, why do you assume $T\geq \tilde{O}(m+n)$ (first) and then $T=\tilde{O}(m+n)$ (also as in the statement of the lemma) rather than $T\geq O(m+n)$? Firstly, the result is only interesting if the condition is a lower bound on $T$ only (I am confident this holds but it is not stated as such. Secondly and more importantly, it doesn't seem like the tilde is required since there are no log terms. In particular, if we interpret the tilde notation as possibly involving dividing factors of log terms (since we want a lower bound on $T$ not an upper bound),  the second to last inequality on the sequence of inequalities in line 788.5 could be incorrect (though the last inequality with $\tilde{O}(\alpha \sqrt{T})$ remains correct where the log terms are $\log(mn)$.


**4** Could you include a more detailed comparison to the existing results in [1] and [4]?




**References**

[1] Elad Hazan, Satyen Kale, and Shai Shalev-Shwartz, "Near-optimal algorithms for online matrix prediction", COLT 2012.

[2] Elad Hazan et al. "Introduction to online convex optimization." Foundations and trends in optimization, 2016.

[3] Nathan Srebro, "Rank, Trace-Norm and Max-Norm", COLT 2005.

[4] Prateek Jain, Soumyabrata Pal Online Low Rank Matrix Completion. ICLR 2023.

[5]  Yuxin Chen, Yuejie Chi, Jianqing Fan, Cong Ma, Yuling Yan. Noisy Matrix Completion: Understanding Statistical Guarantees for Convex Relaxation via Nonconvex Optimization.  SIAM journal of Optimization, 2020.

**Limitations:**

1. The experiments section is very preliminary. No attempt is made to evaluate the online learning setting or to compare with other baselines. This is also an extremely small artificial dataset most likely constructed to be able to run experiments very fast on a laptop.

2. See "weaknesses".

---

> ### Author Rebuttal · Authors · 2023-08-09
>
> Thank you very much for your very detailed review of our work!
>
> Weaknesses:
>
> 1.  Lemma 3 in Appendix A1: Thank you for pointing out the mistake. We will change this. In fact, the convexity of the constraint set follows directly from the (non-trivial) fact that max-norm is a well-defined norm.
>
> 2. The explanations are not always perfectly self-contained and assumptions are not always well introduced.
>
> 2.a: Thank you for pointing this out. Yes, we should remove the $\tilde{C}$.
>
> 2.b: Thank you for pointing this out. There is a mistake in  the preamble. Corollary 21 holds for arbitrary distributions. We will change the writing accordingly.
>
>
> 3. Thank you for your suggestions. We will improve our writing. Matrix entropy is used in Algorithm 5.
>
> All minor issues and typos we will fix accordingly. Thank you very much for your very detailed feedback.
>
> Questions:
>
> 1.a I don't understand why problem (7) is equivalent to problem MP 3. Indeed, I can't see why the PSD condition appears. I have a feeling this comes from the $(\beta, \tau)$ decomposability, but there is definitely an argument missing.
>
> **Response**: Indeed it is not immediately equivalent, and there are subtleties here. Thanks for pointing this out. We made changes to the appendix accordingly.  In fact, we could show that with added symmetric PSD condition, the problem is NP-hard. We add this as only an indication of computational hardness in a special case. We will clarify this in the appendix of the final version.
>
> 1.b The first argument in the "soundness" part which states that we can assume is rank one w.l.o.g. I also don't understand this argument: why is it the case that the matrix achieves value if the matrix  does. This is absolutely not obvious to me. I agree with the rest of the proof assuming this is correct.
>
> **Response**: The proof shows that given a solution, we can construct a rank-1 solution with the claimed objective value, which is sufficient for our proof. The construction is simply taking the rank-1 matrix ww^T for w_i = root(X_{ii}). The objective value of this rank-1 matrix is at least the objective value of $X$ since $X_{ii}X_{jj}\geq X_{ij}^2$ for symmetric psd $X$.
>
> 2. Can you clarify whether the expectation in the equation on line 603.5 (bottom of page 18) refers only to the randomness in the choice of the loss function $h_t$ (which is assumed to be "stochastically i.i.d.). In that case, shouldn't there also be an expectation on the right hand side in front of the regrets?
>
> **Response**: Indeed. Here the regret is the expected regret. We will make sure to clarify this.
>
> 2.2 The lengthy discussion at the beginning of Section B culminating in the equation from line 603.5 seems to mostly justify the validity of the Game-Regret (see line 679) as a measure of performance of the algorithm by linking it to the minimax regret. Why not apply the result from line 603.5 and make a self-contained theorem which applies to the minimax regret in the paper's setting?
>
> **Response**: Line 603.5 holds for bilinear shared objectives. The shared objective function in the online PMC setting is not bilinear. However, it still motivates the use of Game-Regret as a measure of performance (also indicated by the Corollary 21).
>
> 3 On lines 786-788, why do you assume $T\geq \tilde{O}(m+n)$ (first) and then $T= \tilde{O}(m+n)$ (also as in the statement of the lemma) rather than $T\geq \tilde{O}(m+n)$? Firstly, the result is only interesting if the condition is a lower bound on $T$ only (I am confident this holds but it is not stated as such. Secondly and more importantly, it doesn't seem like the tilde is required since there are no log terms. In particular, if we interpret the tilde notation as possibly involving dividing factors of log terms (since we want a lower bound on $T$ not an upper bound), the second to last inequality on the sequence of inequalities in line 788.5 could be incorrect (though the last inequality with $\tilde{O}(\alpha\sqrt{T})$ remains correct where the log terms are $\log(mn)$.
>
> **Response**:
> Thanks for pointing this out. First of all, tilde is not needed, and we are upper bounding $T$ here. The more accurate statement is that the LHS is bounded by Te^{-\beta}mn. The range of parameters we are interested in, is T ~ poly(mn). We can therefore choose the constants in \beta to be as we need. We will clarify this in the later version. All we are trying to say is that by restricting to the constrained simplex, we do not lose much in the optimal point of the objective. For every $t$, the objective functions of the two $C$’s differ by order of magnitude $1/mn$. When summing over $T$ iterations, the difference between the two optimums is bounded by $\alpha T/mn$ (constant omitted). Since to achieve $\epsilon$-error in regret, we only need $T$ to be of magnitude linear (up to logarithmic terms) of dimension $m+n$, this difference is small. In fact, since our regret is dominated by $\sqrt{T}$, we might only need this sum of differences to be bounded by $\sqrt{T}$, i.e. $T/mn \leq \sqrt{T}$, which is clearly true as we at most need to see all $mn$ entries to complete the matrix.
>
> 4 Could you include a more detailed comparison to the existing results in [1] and [4]?
>
> **Response**: We are happy to do it in the full version. The main difference is that we are tackling a partial completion problem, whereas these papers consider full completion. However, we make extensive use of their techniques, especially in the online algorithm and its analysis. Notice that, however, we have a different primal-dual definition of regret. We are happy to elaborate more in the full version.

---

> > ### Comment · Reviewer_wLHP · 2023-08-18
> > **Thanks+ follow up**
> >
> > Many thanks for the clarifications. Thank you, in particular, for agreeing to write a correct proof of Lemma 3 and to fix the minor issues in the Lemma 25.
> >
> > Also, thanks for clarifying my doubts regarding the first argument in the soundness part. That was my bad, it was reasonably understandable in the first draft, actually.
> >
> > On the other hand, there are two points where your answer is satisfactory but I would really like you to include a better explanation in the paper to avoid any confusion. Those are:
> > 1. The fact that MP3 is not, in fact equivalent to quation (7). In addition, could you also clarify (in this rebuttal and in the paper), exactly what you mean by "we could show that with added symmetric PSD condition, the problem is NP-hard". Do you mean that simply adding a symmetric PSD condition makes the problems equivalent? If so, why?
> >
> > 2. The fact that since line 603.5 only holds for bilinear objectives, the section on the definition of the game-regret should be taken as a motivation but cannot be used to derive a bound on minimal regret (since in the current form, it may seem that you are claiming a minimax regret bound).
> >
> > Other than the concerns above, I am still convinced of the very high quality of this paper. Congratulations!

---

### Official Review · Reviewer_PBAT · 2023-07-02

**Soundness:** 3 good
**Presentation:** 3 good
**Contribution:** 4 excellent
**Rating:** 7
**Confidence:** 4

**Summary:**

Typical matrix completion methods aim to recover the whole matrix, based on strong conditions on the matrix itself as well as the sampling distribution. In contrast, this paper proposes a method to complete a subset of the entries with high confidence, which can bypass the need of these conditions. More specifically, the proposed approach builds on top of existing matrix completion methods, and identifies which completed entries can be recovered well. A computational efficient algorithm (as well as an inefficient one) is proposed. Corresponding theoretical guarantees are also provided.

**Strengths:**

- This paper studies an important and interesting problem, as certain necessary conditions (on matrix structure and sampling distribution) required for matrix recovery are not always guaranteed to hold. In such ill-posed settings, partial guarantee is useful. I am not aware of any prior work for partial matrix completion, so it is a novel contribution to this area.
- This work provides strong theoretical guarantee for the proposed methods.

**Weaknesses:**

The authors define the "Partial Matrix Completion Problem" (Line 223) as finding a specific $C$; but the theoretical guarantee does not seem to align directly.

**Questions:**

- Corollary 2 (Section 1.1): is this $|C|$ the cardinality or the sum as defined later in Section 2.3?
- The explanation after "The Partial Matrix Completion Problem" is unclear. In line 227, what is $C$ ? Based on the explanation, I think we can guarantee that for $C=\mu$ , but I am not sure if it works for all $C$ .
- The authors specify "the partial matrix completion problem" in line 223 as finding a *specific* $C$ . I am curious if the proposed algorithms has any guarantee about finding such $C$ defined in 223. Note that the theoretical results (Theorems 8 and 10) provide guarantee in terms of $\frac{1}{C}\sum_{x\in\mathcal{X}}C_x(\hat{M}_x - M_x^\star)^2$ . This type of guarantee looks good to me. I just want to see the relationship with the *target* as specified in "The Partial Matrix Completion Problem".
- How does misspecification in rank (in the MC step) affect the performance  of the proposed method?
- How to choose the tuning parameters in practice?

**Limitations:**

The authors do not provide an explicit discussion on the limitations of their work. I think more extensive numerical experiments would improve this work. I am particular interested in how the proposed methods perform under different sampling distributions (ranging from some easy settings where completion of the whole matrix is possible, to ill-posed settings.)

---

> ### Author Rebuttal · Authors · 2023-08-09
>
> Thank you for taking the time and efforts in reviewing our work! Here are the resonses to your concerns:
>
> Weaknesses:
>
> 1. The authors define the "Partial Matrix Completion Problem" (Line 223) as finding a specific $C$; but the theoretical guarantee does not seem to align directly.
>
> **Response**: Our definition is not saying that there is a unique $C$ in particular which satisfies the problem. Hence, we find such a $C$ which is sufficiently good. Although our result is not strictly maximizing $|C|$, we provide a meaningful lower bound guarantee. Later in the online section we even give an agnostic guarantee: we compete with the best C, even though such a (optimal) C may have high loss. This is common in online and agnostic learning.
>
> Questions:
>
> 1. Corollary 2 (Section 1.1): is this $|C|$ the cardinality or the sum as defined later in Section 2.3?
>
> **Response**: Note that in Section 1.1, these two are the same as $C\in \\{0,1\\}^{m\times n}$. We will make sure to clarify this.
>
> 2. The explanation after "The Partial Matrix Completion Problem" is unclear. In line 227, what is $C$? Based on the explanation, I think we can guarantee that for $C=\mu$, but I am not sure if it works for all $C$.
>
> **Response**: $C$ in this case is the solution to problem (2) in line 223. The guarantees are spelled out in Theorem 8 and 10. This is just a summary of the result. We will make sure to clarify this.
>
> 3. The authors specify "the partial matrix completion problem" in line 223 as finding a specific $C$. I am curious if the proposed algorithms has any guarantee about finding such $C$ defined in 223. Note that the theoretical results (Theorems 8 and 10) provide guarantee in terms of
>
> $\frac{1}{C} \sum_{x \in {\cal X} } C_x (\hat{M}_{x} - M^*_{x})^2$.
>
> This type of guarantee looks good to me. I just want to see the relationship with the target as specified in "The Partial Matrix Completion Problem".
>
> **Response**: Note that  $\frac{1}{C} \sum_{x \in {\cal X}} C_x (\hat{M}_{x}-M^*_{x})^2$ is exactly the constraint in line 223 defined in line 218. Yet it is interesting to explore other types of guarantees.
>
> 4. How does misspecification in rank (in the MC step) affect the performance of the proposed method?
>
> **Response**: While we haven’t explored this, our conjecture is that the theoretical guarantee should translate in the natural way (scale with the rank/complexity measure scale in the guarantee). In practice, we have not explored this yet. It is interesting to explore this direction.
>
>
> 5. How to choose the tuning parameters in practice?
>
> **Response**: Possible tuning methods include hyperparameter sweep, and it is possible that more sophisticated methods can help, such as hypergradient descent and meta optimization.
>
> Limitations:
>
> The authors do not provide an explicit discussion on the limitations of their work. I think more extensive numerical experiments would improve this work. I am particular interested in how the proposed methods perform under different sampling distributions (ranging from some easy settings where completion of the whole matrix is possible, to ill-posed settings.)
>
> **Response**: We agree and thank the reviewer for  this suggestion—we will attempt to do this by the final version of the paper (or in future work, if that’s needed).

---

> > ### Comment · Reviewer_PBAT · 2023-08-17
> >
> > Thank you for your response. Thanks for pointing out the $C$ defined by "Partial Matrix Completion Problem" (Line 223) might not be unique. My original comment focuses on that the target in Line 223 is defined as those that **maximizes** $|C|$ under the constraint. So I think that one natural measure of convergence would be based on some distance to this set (of C that maximize $|C|$ under the constraint). But I agree that the results obtained in this work are meaningful.

---

### Official Review · Reviewer_n6bT · 2023-07-04

**Soundness:** 4 excellent
**Presentation:** 3 good
**Contribution:** 4 excellent
**Rating:** 7
**Confidence:** 4

**Summary:**

This paper considers a twist on the standard matrix completion problem where one is required to only
complete a subset of entries (not the whole matrix) that includes the entries shown. This allows them to consider
substantially more observation patterns unlike the standard missing-at-random response. In this context
the paper makes the following contributions:

1. Propose a computationally-inefficient algorithm that with high probability recovers a subset of entries
that is at least as large as the revealed set, with low target accuracy.
2. Develop a computationally-efficient relaxation of this algorithm that has worse statistical dependence
on the target accuracy than the inefficient algorithm.
3. Provide an online variant of the algorithm that is iterative/gradient based that applies to adversarial
online matrix completion.


**Strengths:**

In my view the main strengths of the paper are:

1. Posing an interesting partial matrix completion setting
2. Leveraging existing matrix completion work and separating the problem of matrix completion from that of obtaining good coverage (possibly fractional)
3. Generalizing to the online and adversarial setting



**Weaknesses:**

Some weaknesses, though these are better understood as interesting avenues for future work:

1. The proposed algorithms provide noisy completion even when the observations are noiseless (i.e. no obvious notion of completion to the inherent noise level).
This does seem inherent to the algorithm/proof techniques used here, it is unclear to me that this can be overcome here.
2. There is an obvious sample complexity gap between the efficient and inefficient algorithms. Is this inherent, or a result of the proof technique here?




**Questions:**

1. The algorithmic viewpoint here is to separate the coverage computation from the completion method. This is advantageous in some ways (notably the generality and e.g. obtaining an essentially free proof of the risk), but potentially disallows using structure in the completion. E.g. in the causal inference setting of (say) row $1$ being partially revealed from columns $1, 2\ldots, K \leq \text{dim}(M)$.
2. What is $\mu_\max$? It is defined in the appendix (and easy to guess) but is not in the text (unless I missed)


**Limitations:**

No significant limitations, addressed previously.

---

> ### Author Rebuttal · Authors · 2023-08-08
>
> Thank you for taking the time and efforts in reading and reviewing our work! Here are our responses to your concerns:
>
> Weaknesses:
>
> 1. The proposed algorithms provide noisy completion even when the observations are noiseless (i.e. no obvious notion of completion to the inherent noise level). This does seem inherent to the algorithm/proof techniques used here, it is unclear to me that this can be overcome here.
>
> **Response**: This is correct. It is currently inherent to our algorithm and proof techniques.  It is also interesting to us to explore whether we can have an algorithm that completes missing entries without noise when the observations are noiseless.
>
>
> 2. There is an obvious sample complexity gap between the efficient and inefficient algorithms. Is this inherent, or a result of the proof technique here?
>
> **Response**: Great point, yes: we conjecture that the computational hardness of the nonconvex formulation is inherent, and it may be computationally hard to get the optimal statistical complexity. Such results are known in, for example, sparse recovery (LASSO), and it would be very interesting to research this direction further.
>
> In general, yes, there is a lot more investigation to be done here, and these are good points. We believe we have only initialized research in this direction. We’ll add it to the future works section.
>
> Questions:
>
> 1. The algorithmic viewpoint here is to separate the coverage computation from the completion method. This is advantageous in some ways (notably the generality and e.g. obtaining an essentially free proof of the risk), but potentially disallows using structure in the completion. E.g. in the causal inference setting of (say) row $1$ being partially revealed from columns $1,2,\dots,K\leq\mathrm{dim}(M)$.
>
> **Response**: Note that the offline algorithms do separate the coverage computation from the completion method, but the online algorithm does not as clearly do this through online computation of a version space for the completion. Thus, in effect we are showing that both can be done.
>
> 2. What is $\mu_{\max}$? It is defined in the appendix (and easy to guess) but is not in the text (unless I missed).
>
> **Response**: Yes, $\mu_{\max}$ is the maximum entry over all entries in $\mu$. We will add this definition earlier.

---

> > ### Comment · Reviewer_n6bT · 2023-08-21
> > **Thanks for the response**
> >
> > I'll first thank the authors for their response to mine and other reviewers' comments. They have addressed my concerns. Hope to see the paper published soon.

---

### Official Review · Reviewer_Hbwc · 2023-07-07

**Soundness:** 3 good
**Presentation:** 2 fair
**Contribution:** 3 good
**Rating:** 5
**Confidence:** 3

**Summary:**

The authors discuss the Partial Matrix completion and present an algorithm for completing a large subset of the entries with high confidence. They also aim to find the number of entries that can be completed (coverage) with a small error where the samples are from an unknown sampling distribution.

**Strengths:**

- Identifying the entries which have high confidence without the incoherence assumption is an important problem and the problem is described well by the authors, including the relevant citations.

- Two alternative methods are presented. An inefficient (but stronger) algorithm, as well as an efficient (but worse sampling bounds), for finding the optimal coverage for partial matrix completion. Their limitations and disadvantages are discussed honestly. (i.e. The inefficient algorithm requires solving a quadratic function as an optimization function which is also shown to be NP-hard. )



**Weaknesses:**

- The online formulation of the "partial matrix completion" is said to be one of the main contributions of the paper however it is only discussed in the appendix. It would be nice to see the main contribution in the main paper.
- Experimental support is missing significantly. The experiments can be enriched to support the claims of the authors, further baselines are missing. Time analysis is missing. Performance could be highlighted and leveraged more.
- The guarantees are discussed before introducing the algorithms.
- The definition of "Partial" could have been emphasized more, since the paper is directly built on that term.

Minor :

- The Figures are not numbered and not addressed in detail.
- The organization of the paper could be further adjusted. Some terms are used before they were defined, the styling is not appropriate which makes it difficult to follow the paper. i.e. Algorithm 1, "FullComp", or the styling of the function (3).

**Questions:**

1. How did the authors select the data? How did they decide which user and which items to consider? Which exact Movie Lens dataset is used? There are multiple versions of the same dataset. Some further clarification would be appreciated.

2. What are some practical domains that the presented algorithms can be useful? How is the time complexity reflected in the larger experimental settings? How much time does it take for your algorithm to run to find the coverage?

**Limitations:**

No direct negative societal impact exists.

---

> ### Author Rebuttal · Authors · 2023-08-09
>
> Thank you very much for taking the time to read and review our paper. We note that despite the score you gave us, your comments are generally positive, and you recognize the contribution and soundness of our work. Out of the four weaknesses you pointed out, three seem to be stylistic issues, and we are happy to try to improve the presentation based on your suggestions. We will address your concerns and hopefully these responses can give you a chance to reevaluate our work!
>
>
> Weaknesses:
>
> Major:
>
> 1. The online formulation of the "partial matrix completion" is said to be one of the main contributions of the paper however it is only discussed in the appendix. It would be nice to see the main contribution in the main paper.
>
> **Response**: if the paper is accepted, we will have an additional page for the camera ready version which we will use to discuss the online version in the paper body.
>
> 2. Experimental support is missing significantly. The experiments can be enriched to support the claims of the authors, further baselines are missing. Time analysis is missing. Performance could be highlighted and leveraged more.
>
> **Response**: Our contribution is mainly conceptual and theoretical, and therefore the experimental section was added to elucidate and support the framework and theory, rather than to give an extensive evaluation or comparison as in applied papers. Significantly more experiments would thus be out of scope, but we are very happy to compare to other methods (we don’t know any since it’s a new framework, please refer us if you have anything in mind), or add more experiments that would help explain the framework, can you please suggest such? We want to emphasize that there exists no baseline we can compare to as partial matrix completion is a completely new framework, which is part of the novelty of our work. This is also why we use a standard matrix completion package to compare the squared error of that package with the test values with respect to our obtained confidence matrix. We are not sure what you mean by time analysis.
> Can you elaborate on an experiment that would help the reader understand the main concepts we introduce? We would be happy to add any other evaluation, time permitting (space limitations would mean this would go in the appendix in any case, since we have so many other contributions in the paper).
>
> 3. The guarantees are discussed before introducing the algorithms.
>
> **Response**: Thank you for the suggestion. We chose to qualitatively discuss the guarantees first because we felt it was more intuitive. It is generally common to present the guarantees of an algorithm before its implementation details, however, if you feel that the inputs or the setting is not clear early enough, please let us know.
>
> 4. The definition of "Partial" could have been emphasized more, since the paper is directly built on that term.
>
> **Response**: Thank you for the suggestion. We have added a sentence emphasizing this in the introduction.
>
>
> **Response to minor weaknesses**: Thank you for your suggestions. We will change according to your suggestions.
>
> Questions:
>
> 1. How did the authors select the data? How did they decide which user and which items to consider? Which exact Movie Lens dataset is used? There are multiple versions of the same dataset. Some further clarification would be appreciated.
>
> **Response**: We use the MovieLens 100K dataset. We use a 250x250 submatrix from the 943x1682 matrix. We chose to select the MovieLens dataset because it is a classic dataset used for matrix completion problems. We use a simplified version of the ODD as outlined in Appendix D.
>
> 2. What are some practical domains that the presented algorithms can be useful? How is the time complexity reflected in the larger experimental settings? How much time does it take for your algorithm to run to find the coverage?
>
> **Response**: While our results are mainly theoretical, we believe they have meaningful applications as well. Specifically, the online algorithm is designed to be efficient and can be used to evaluate to what extent entries can reliably be completed in a matrix, which is an important problem in any scenario where decisions should only be made if there is high confidence. For example, in datasets which pertain to hospital patients and their features, one would only want to make predictions for patients which can be made with high confidence. Even in low stakes settings, such as movie recommendations, companies, such as Netflix seem to be already abstaining from making predictions on some movies today, as discussed in the paper. This is only a first step and we are hoping later papers will find even more applications.
>
>
>
> In light of our changes and rebuttal, would you consider adjusting your score? We are more than happy to answer any further questions.

---

> > ### Comment · Reviewer_Hbwc · 2023-08-14
> >
> > My main concern was about the experimental support provided in the paper.
> >
> > After carefully reading the comments of other reviewers' comments as well as the rebuttal comments of the authors, I changed my evaluation to borderline acceptance.
> >
> > I agree with the points mentioned by the reviewer wLHP as weaknesses.
> >
> > Thanks.

---

### Decision · Program_Chairs · 2023-09-21

**Decision:**

Accept (poster)

**Comment:**

The reviewers are unanimous that this interesting twist on the matrix completion problem deserves publication. The reviews are high quality and demonstrate detailed understanding of the paper, and the discussion was vigorous.